Discovering unknown Madagascar biodiversity: integrative taxonomy of raft spiders (Pisauridae: Dolomedes)

Yu Kuang-Ping 1 2 Kuang-Ping.Yu@nib.si
http://orcid.org/0000-0002-0057-2178 Kuntner Matjaž 1 3 4 5
1 Department of Organisms and Ecosystems Research, National Institute of Biology , Ljubljana , Slovenia
2 Department of Biology, Biotechnical Faculty, University of Ljubljana , Ljubljana , Slovenia
3 Jovan Hadži Institute of Biology, ZRC SAZU , Ljubljana , Slovenia
4 Department of Entomology, National Museum of Natural History, Smithsonian Institution , Washington, D.C. , United States of America
5 State Key Laboratory of Biocatalysis and Enzyme Engineering, and Centre for Behavioural Ecology and Evolution, School of Life Sciences, Hubei University , Wuhan, Hubei , China
Sierwald Petra
Electronic publication date: 2024 Feb 27
Publication date: 2024
Volume: 12
Electronic Location ID: e16781
Received 2023 Oct 4; Accepted 2023 Dec 18
Copyright: © 2024 Yu and Kuntner
Copyright year: 2024
Copyright holder: Yu and Kuntner
License: This is an open access article distributed under the terms of the Creative Commons Attribution License, which permits unrestricted use, distribution, reproduction and adaptation in any medium and for any purpose provided that it is properly attributed. For attribution, the original author(s), title, publication source (PeerJ) and either DOI or URL of the article must be cited.
License URL: https://creativecommons.org/licenses/by/4.0/

Keywords: Fishing spiders, Endemic species, Species delimitation, Integrative taxonomic model, Unified species concept

Funding: Slovenian Research and Innovation Agency P1-0255 J1-9163 Matjaž Kuntner and Kuang-Ping Yu were supported by the Slovenian Research and Innovation Agency (grants P1-0255; J1-9163). The funders had no role in study design, data collection and analysis, decision to publish, or preparation of the manuscript.

==============================
Madagascar is a global biodiversity hotspot, but its biodiversity continues to be underestimated and understudied. Of raft spiders, genus Dolomedes Latreille, 1804, literature only reports two species on Madagascar. Our single expedition to humid forests of eastern and northern Madagascar, however, yielded a series of Dolomedes exemplars representing both sexes of five morphospecies. To avoid only using morphological diagnostics, we devised and tested an integrative taxonomic model for Dolomedes based on the unified species concept. The model first determines morphospecies within a morphometrics framework, then tests their validity via species delimitation using COI. It then incorporates habitat preferences, geological barriers, and dispersal related traits to form hypotheses about gene flow limitations. Our results reveal four new Dolomedes species that we describe from both sexes as Dolomedes gregoric sp. nov., D. bedjanic sp. nov., D. hydatostella sp. nov., and D. rotundus sp. nov. The range of D. kalanoro Silva & Griswold, 2013, now also known from both sexes, is expanded to eastern Madagascar. By increasing the known raft spider diversity from one valid species to five, our results merely scratch the surface of the true Dolomedes species diversity on Madagascar. Our integrative taxonomic model provides the framework for future revisions of raft spiders anywhere.

Introduction

Madagascar is well known for its diverse and unique biota (Antonelli et al., 2022). The island’s diverse habitats, including humid forests, dry spiny forests, tapia woodlands, and coastal areas, provide niches for an incredible variety of flora and fauna found nowhere else on the planet. Madagascar is renowned for its iconic inhabitants, such as lemurs, chameleons, and baobab trees, but the richness and endemicity extend to numerous other organisms, including insects, reptiles, birds, and spiders. Although currently positioned close to continental Africa, Madagascar was part of the Indian subcontinent for around 50 million years since the break off from Gondwana (Sanmartín & Ronquist, 2004; Ali & Aitchison, 2008). The combination of a unique geological history of Madagascar and its long isolation for around 84 million years has affected the evolution of organisms inhabiting the island resulting in extremely high endemism rates (Buerki et al., 2013; Antonelli et al., 2022). Despite a long and lively history of biodiversity research of Madagascar its biota continues to be vastly underestimated. Bridging this knowledge gap is critical for conservation of Madagascar’s biodiversity that is undergoing major threats and losses due to human activities (Ralimanana et al., 2022).

The globally distributed spider genus Dolomedes Latreille, 1804—known as raft-or fishing spiders—contains around 100 species (World Spider Catalog, 2023). Being large and iconic, with semi-aquatic lifestyles, and often predators of freshwater vertebrates in addition to invertebrates, Dolomedes species are model organisms in diverse fields such as behavioral ecology (Bleckmann & Bender, 1987; Suter & Gruenwald, 2000; Johnson, 2001; Johnson & Sih, 2007; Schwartz, Wagner & Hebets, 2013; Kralj-Fišer et al., 2016) and conservation biology (Smith, 2000; Duffey, 2012; Leroy et al., 2013). While Dolomedes has been well surveyed in some regions (Carico, 1973; Smith, 2000; Zhang, Zhu & Song, 2004; Tanikawa & Miyashita, 2008; Vink & Dupérré, 2010; Raven & Hebron, 2018), our knowledge of Dolomedes in the tropics is much more limited. One region that stands out in this respect is Madagascar.

Strand (1907)’s description of Dolomedes saccalavus from Nosy Be (Nossibé) was the first Dolomedes record in Madagascar. However, Silva & Griswold (2013a) considered the species as nomen dubium as the original description was based on a subadult female and the type specimen was destroyed. Dolomedes kalanoro Silva & Griswold, 2013 is currently the only valid Dolomedes species from Madagascar. Silva & Griswold (2013a) described the species from two male specimens collected from rivers in dry and subhumid forests in the west and the south of the island. Although the humid forests in the east and the north of Madagascar harbor the highest biodiversity on the island (Antonelli et al., 2022), surprisingly, no Dolomedes species have been described there. During our single expedition to two national parks, Marojejy in the north and Andasibe-Mantadia in the east of Madagascar, we discovered several morphospecies of Dolomedes. These exemplars have proven to be difficult to assign to species using conventional morphological diagnostics.

As in numerous other clades, defining clear species boundaries among closely related Dolomedes species is difficult using a single line of evidence (see Tanikawa & Miyashita, 2008; Vink & Dupérré, 2010). Recent taxonomic reviews of Dolomedes (Tanikawa, 2003; Zhang, Zhu & Song, 2004; Tanikawa & Miyashita, 2008; Vink & Dupérré, 2010; Raven & Hebron, 2018) have used different combinations of characteristics for species diagnostics (Table 1). The characteristic that all prior studies had in common relate to habitus coloration, the morphology of the male palpal retrolateral tibial apophysis (RTA), and the female epigynal middle field (MF) (Table 1). These structures, however, can show intraspecific variation (see Tanikawa & Miyashita, 2008; Vink & Dupérré, 2010) and may introduce ambiguity in species taxonomy (see Zhang, Zhu & Song, 2004; Tanikawa & Miyashita, 2008). To avoid this problem and to strengthen species boundaries, only two prior studies have added other types of evidence such as DNA barcodes and habitat preferences (Table 1) (Tanikawa & Miyashita, 2008; Vink & Dupérré, 2010). So far, these two are the only revisions of Dolomedes that have approached the definition of integrative taxonomy (Dayrat, 2005).

Table 1 Different types of evidence used (labeled as “Y”) to diagnose Dolomedes species in recent regional reviews and in this article.

Literature	Region	Morphology	Molecular	Ecology	
Somatic	Male palp	Female epigynum	
Coloration	Leg I	Total length	Tibia length	Cy length	RTA	BCA	T	DTP	Sa	MA	Fu	Eb	LA	Margin	MF/EF	Vulva	COI	Actin 5C	Habitat	
Zhang, Zhu & Song (2004)	China	Y					Y	Y			Y	Y	Y			Y	Y	Y				
Tanikawa (2003)	Japan (Ryukyu)	Y		Y	Y	Y	Y										Y	Y				
Tanikawa & Miyashita (2008)	Japan	Y	Y		Y		Y									Y	Y		Y		Y	
Vink & Dupérré (2010)	New Zealand	Y					Y									Y	Y		Y	Y	Y	
Raven & Hebron (2018)	Oceania	Y		Y	Y	Y	Y		Y	Y		Y				Y	Y	Y				
This article	Madagascar	Y	Y	Y	Y	Y					Y	Y	Y	Y	Y	Y	Y	Y		Y	
Note:

BCA, basal cymbium apophysis; Cy, cymbium; DTP, distal tegular projection; Eb, embolus; EF, epigynal fold; Fu, fulcrum; LA, lateral subterminal apophysis; MA, median apophysis; MF, middle field; RTA, retrolateral tibial apophysis; Sa, saddle; T, tegulum.

To establish a comparative framework for future Dolomedes taxonomic discoveries (see Schlick-Steiner et al., 2010; Bond et al., 2022), our study aims to: i) treat variation in morphological features within a statistical framework in order to utilize the best combination of the diagnostic characteristics among Madagascar Dolomedes; and ii) define robust species boundaries among Madagascar Dolomedes using an integrative taxonomic model which includes morphological, molecular, and ecological evidence.

Materials and Methods

Taxon sampling

We collected Dolomedes exemplars by hand both day and night (research permit issued by Direction des Aires Protégées, des Ressources Naturelles renouvelables et des Ecosystèmes; N°166/20/MEDD/SG/DGGE/DAPRNE/SCBE.Re). Specimens were preserved in 75% ethanol for morphological examination. Two to four legs of each specimen were removed and preserved separately in 96% ethanol for DNA extraction and molecular analyses. Specimens examined in this study are held at the National Institute of Biology (NIB) in Slovenia (voucher code KPARA) while the type series are deposited at the National Museum of Natural History, Smithsonian Institution (USNM), Washington, DC, USA (voucher code USNMENT). We also included original Madagascar sequences of a Dolomedes species that is being described from La Réunion (G Cazanove, K-P Yu, B Derepas, A Henrard, 2023, unpublished data). In addition, we examined relevant Dolomedes collections deposited at the Royal Museum for Central Africa, Tervuren, Belgium (RMCA) and the Senckenberg Natural History Museum, Frankfurt, Germany (SMF). Representing an outgroup clade to the Madagascar Dolomedes, we included a sequence of Dolomedes raptor Bösenberg & Strand, 1906 from Taiwan (K-P Yu, K Matjaž, 2022, unpublished data).

Anatomical examination

For morphological examination, measurement, and imaging, we combined use of a classical stereomicroscope (Leica M205C; Leica, Wetzlar, Germany) and a digital microscope (Keyence VHX7000; Keyence, Osaka, Japan). All the measurements given are in millimeters (mm). The measurements of palps consisted of femur, patella, tibia, and tarsus or cymbium while those of legs consisted of femur, patella, tibia, metatarsus, and tarsus. Variation values were presented as “mean ± s.d.”. Female epigyna were dissected and cleaned in Potassium hydroxide (KOH) solution. Male right palps were expanded by repeatedly soaking in distilled water and KOH solution for further morphological analyses.

Integrative taxonomy

Our integrative taxonomy model combined original morphological, molecular, and ecological data (Fig. 1) based on the unified species concept (de Queiroz, 2005, 2007; see also Schlick-Steiner et al., 2010). Using morphological data as the primary evidence (morphology-first, see Hedin & Milne, 2023), we first classified all specimens into groups by tentative species diagnosis. The observed differences and those commonly used diagnostic characteristics (Table 1) were put into detailed size comparisons and intragroup morphometric framework to facilitate species hypotheses. We then tested the species hypotheses using a molecular phylogeny and molecular species delimitation. We also included habitat preferences, potential geological barriers, and dispersal-related traits in adding credibility to the species hypotheses considering potential geneflow limitations.

Figure 1 Integrative taxonomic model applied in this study.

Morphological comparisons and dispersal-related traits

We compared the variation of sizes and shapes of the selected characteristics. For size comparisons, we chose measurements of six characteristics: carapace width, relative length of leg I (leg I length divided by carapace width), relative length of tarsus I (tarsus I length divided by leg I length), relative length of male palp (palp length divided by carapace width), relative length of male palp Cy (Cy length divided by palp tibia length), and diameter of the male embolic ring (De, see Fig. S1A). We used Photoshop 2022 (Adobe Inc., 2019) to recalibrate the images of male DST into one magnification. We then rotated the images until the tip and the outer basal point of the embolus were positioned on the same horizontal reference line (baseline). A vertical and a horizontal reference line were then added to define the maximum distance between the baseline and the outer margin of the embolic ring. Such maximum distance was given as De (see Fig. S1A). De was then measured under the software ImageJ (Schneider, Rasband & Eliceiri, 2012). We performed one-way analyses of variance (ANOVA) with hypothesized morphospecies as the factor. If the results from the ANOVA were significant (p-value < 0.05), Tukey’s honestly significant difference (Tukey HSD) was followed to test whether the sizes of the selected structures are different between pairs of morphospecies. We performed the above analyses in R version 4.2.1 (R Core Team, 2022).

For shape comparisons, we chose ten structures to be included in morphometric analyses, including three structures form the expanded male right palp, Eb (retrolateral view, Fig. S1A), Fu (retrolateral view, Fig. S1B), and LA (retrolateral view, Fig. S1C); four structures from the male left palp, MA (ventral view, Fig. S2A), Eb (ventral view, Fig. S2B), Fu (ventral view, Fig. S2C), and RTA (posterolateral view, Fig. S2D); and three female genital structures, epigynal margin (ventral view, Fig. S3A), MF (ventral view, Fig. S3B), and vulva (dorsal view, Fig. S3C); see also Table S1 for descriptions of each landmarks. We then recalibrated the images into the same magnification and resized them into 30 cm × 20 cm, 72 dpi in Photoshop 2022. Six additional reference lines (see Fig. S1A) were added to the images of male DST, on the basis of those used in measuring De, to support the consistency of landmark plotting within embolus. We first added the six reference lines at the same position as the baseline. With the intersection point between the baseline and the vertical line as the center, we then rotated each line to equally divide the embolus by 22.5 degrees.

We imported the images to the software ImageJ for landmark plotting. As landmarks with geometrical features can be clearly pinpointed in most of the structures, most of the landmarks chosen in this study are Type II landmarks determined by the following criteria: i) the maximum or minimum curvatures, ii) the attachment or intersection point between borders, or iii) the tip of the structure. In the expanded Eb of the male right palp, semilandmarks (Bookstein, 1997), determined by equal-angular spaced reference lines, were also used to interpret the round shape. See Table S1 for details of landmarks in each structure. We performed Generalized Procrustes analyses (GPA) to isolate the shape allometries caused by sizes. Each landmark configuration was rotated and scaled by its centroid size. Via GPA, the shape component of the structure of each specimen can be represented by a point projected in an n-dimensional space, in which n equals the number of landmarks. We then performed principal component analyses (PCA) to determine the two component axes that demonstrate the highest proportion of shape variation. For details of this methodology, see Klingenberg (2016).

Because small sample sizes can introduce unwanted sampling error in morphometrics (Cardini & Elton, 2007), we avoided the commonly used one-way multivariate analysis of variance in determining whether the shape components are significantly different among species. Instead, we investigated how shape components change along the first two PC axes (PC1, PC2). Distribution patterns of the projection points on the first two PC axes were used to i) support statistical evidence in facilitating morphospecies hypotheses, ii) compare shape variations of different characteristics within and among morphospecies, and iii) support the species diagnosis in the taxonomy section. All the morphometric analyses were performed in R version 4.2.1 using the package “geomorph” (Adams & Otárola-Castillo, 2013).

We used the relative length of leg I and tarsus I from the above size comparison to estimate the dispersal abilities of each species. Following the basic assumption that the length of legs could refer to the ability of walking, species with relatively long leg I were considered to be good terrestrial dispersers. On the other hand, Dolomedes are known for their rafting behavior and aquatic dispersal (Suter, 1999; Suter, Stratton & Miller, 2004; Duffey, 2012). By linking the morphology and habitat preferences among groups of wandering spiders, Lapinski, Walther & Tschapka (2015) indicated that the relatively longer tarsus may facilitate semi-aquatic locomotion. Therefore, we considered species with relatively long tarsus I to be good aquatic dispersers.

Molecular phylogeny and species delimitation

We extracted genomic DNA of Madagascar Dolomedes by Qiagen DNeasy Blood & Tissue Kit (Qiagen, Hilden, Germany) following the manufacturer’s instructions. We amplified cytochrome c oxidase I (COI) from at least a pair of mature Dolomedes per morphospecies per locality for the reconstruction of gene trees and for molecular species delimitation. PCR reaction mixture (25 μL) contained 12.5 μL of EmeraldAmp MAX HS PCR Master Mix (Takara Bio Inc, USA), 0.5 μL (10 pmol/μL) of the forward primer (LCO: 5′-GGTCAACAAATCATAAAGATATTGG-3′, Folmer et al., 1994) and the reverse primer (Maggie: 5′-GGATAATCAGAATATCGTCGAGG-3′, Hedin & Maddison, 2001), 8.0–9.0 μL of distilled water, and 2.5–3.5 μL of genomic DNA. Sequence amplification protocols started at 94 °C for 2 min followed by 35 cycles of 30 s of denaturation at 94 °C, 30 s of annealing started at 46 °C, +0.3 °C per cycle until 52 °C, and 120 s of polymerizing at 68 °C. PCR products were sent to Macrogen Europe B.V. (Amsterdam, Netherlands) for purification and sequencing. All sequences were edited and aligned in Geneious Pro 5.6.7 and uploaded to GenBank (see Table S2).

We first performed Assemble Species by Automatic Partitioning (ASAP) (Puillandre, Brouillet & Achaz, 2021) under Kimura two genetic distance substitution model (Kimura, 1980). To obtain COI gene trees, we performed a maximum likelihood (ML) and a Bayesian inference (BI) phylogenetic analysis on the CIPRES Science Gateway portal (Miller, Pfeiffer & Schwartz, 2010) with sequences partitioned by codon. The ML analysis was done in RAxML with 1,000 bootstrap replicates using the program’s rapid bootstrapping algorithm with the GTRCAT substitution model (Stamatakis, 2014). The BI analysis was performed in MrBayes (Huelsenbeck & Ronquist, 2001) under the substitution model GTR + I + G, suggested by jModelTest (Posada, 2008), with 10 million generations run independently in two chains. Trees were sampled every 10,000 generations with 25% burn-in. To select a more robust topology and to examine whether both analyses supported each node, we summarized the topologies of the two gene trees by SumTrees Version 4.0.0 (Sukumaran & Holder, 2015) under DendroPy (Sukumaran & Holder, 2010). The topology with overall higher nodal supports was set as the target tree. We then imported the summarized tree into a Bayesian implementation of the Poisson tree process (bPTP) species delimitation under maximum likelihood and Markov chain Monte Carlo (Zhang et al., 2013).

Habitat preferences estimation and geological barriers

Based on our field observations and the literature by Vink & Dupérré (2010), we chose canopy coverage (open or dense) and water velocity (flowing or standing) as the factors in classifying habitats. We then classified the localities where Dolomedes were collected into four categories (Fig. S4) to generate a rough estimation of the habitat preference of each species.

We estimated the geological barriers by the connectivity of rivers or waterbodies between localities where Dolomedes specimens were collected. We obtained the information on river drainages from the maps provided by the two national parks and the open online data sources on DIVA-GIS (https://www.diva-gis.org). Localities were pinpointed to the map using QGIS 3.22.10 (QGIS.org, 2023). We considered those localities connected to the same river drainage to represent only low geological barriers with relatively higher possibilities of gene flow.

Nomenclatural acts

The electronic version of this article in Portable Document Format (PDF) will represent a published work according to the International Commission on Zoological Nomenclature (ICZN), and hence the new names contained in the electronic version are effectively published under that Code from the electronic edition alone. This published work and the nomenclatural acts it contains have been registered in ZooBank, the online registration system for the ICZN. The ZooBank LSIDs (Life Science Identifiers) can be resolved and the associated information viewed through any standard web browser by appending the LSID to the prefix http://zoobank.org/. The LSID for this publication is: urn:lsid:zoobank.org:pub:C9091268-EC61-41CD-A20C-5C7DC08DAD46. The online version of this work is archived and available from the following digital repositories: PeerJ, PubMed Central SCIE and CLOCKSS.

Results

Of the collected 69 Dolomedes individuals in Parc National de Marojejy and Parc National d’Andasibe-Mantadia, Madagascar, 29 were females, 21 were males, and 19 were juveniles. We initially classified these exemplars into two groups, the “Kalanoro” group, and the “Hydatostella” group, based on the habitus coloration (see also Taxonomy) and the measurements of the carapace width (see Figs. 2A, 2B). The female and male carapace width of the “Kalanoro” group exceeded 7 and 6.5 mm, respectively, whereas in the “Hydatostella” group they were below these values. Within the “Kalanoro” group, we identified one morphospecies as D. kalanoro, and hypothesized two additional morphospecies, tentatively named “gregoric”, and “bedjanic” based on the genital anatomy of both sexes (see Taxonomy). Individuals of the “Hydatostella” group were more uniform, but based on constant genital differences (see Taxonomy), we hypothesized two morphospecies, “hydatostella” and “rotundus”.

Figure 2 Size comparisons of the selected somatic characters among the five morphospecies.

(A and B) Carapace width. (C and D) Relative length of leg I. (E and F) Relative length of tarsus I. (G and H) Relative length of palp. Bold line: median; upper margin of the box: first quartile (Q1); lower margin of the box: third quartile (Q3); upper whisker: the maximum or Q1 + 1.5 × interquartile range (IQR); lower whisker: the minimum or Q3 − 1.5 × IQR of each group of data; hollow circle: outlier; n: number of specimen(s).

Morphological comparisons

Somatic characters

Comparisons in the carapace width and relative length of leg I, tarsus I, and palp together support the two-group classification stated above but cannot fully separate the morphospecies within the groups (Fig. 2, see also Tables S3, S4). The morphospecies of the “Kalanoro” group in general have a wider carapace (Figs. 2A, 2B), and longer appendages (Figs. 2C–2H) compared to the “Hydatostella” group. Such differences in leg I and tarsus I also refer to better abilities in walking and aquatic locomotion in the “Kalanoro” group (Figs. 2C–2F). Carapace width of male and female “gregoric” are the only exception failing to fit the two-group classification as they are not significantly wider than the “Hydatostella” group (see Table S4). Only the width of the female carapace (Fig. 2A) and the relative length of the male palp (Fig. 2H) can partially separate the morphospecies of the “Kalanoro” group. The “gregoric” females have a narrower carapace compared to the other two morphospecies of the group (Fig. 2A). The “bedjanic” males have longer palps than D. kalanoro (Fig. 2H).

Male genital characters

The shape of LA (Fig. 3) and the ventral view of the Eb (Figs. 4A, 4B) can well separate males of all morphospecies in both groups. The measurements and the shapes of the other male palpal structures can only partially support the separation of the morphospecies within their group (Figs. 4C and 4D, 5–8). Within the “Kalanoro” group, the three morphospecies are well separated by i) the length (Fig. 4A, PC1) and the width (Fig. 4A, PC1 & 2) of the basal Eb in both the ventral and the retrolateral view after palp expansion (Fig. 4C, PC1); ii) De (Fig. 5A, se also Tables S3, S4); iii) the curvature of the retrolateral edge of the MA (Fig. 6A, PC1 & 2); and iv) the curvature of the posterior (Fig. 3A, PC1) and the dorsal edge (Fig. 3A, PC2) of LA. The relative length of Cy (Fig. 5B, see also Tables S3, S4), shapes of Fu in both views (Fig. 7), and RTA (Fig. 8A) can only help distinguish D. kalanoro from “bedjanic” while “gregoric” can match either of the other two morphospecies. The two morphospecies within the “Hydatostella” group differ in i) curved vs straight retrolateral arc of the Eb in ventral view (Fig. 4B, PC1); ii) the shape and relative size of the dorsal lobe of the RTA (Fig. 8B, PC1); and iii) the curvature of the dorsal edge of the LA (Fig. 3B, PC1). However, De (Fig. 5A, see also Tables S3, S4), relative length of Cy (Fig. 5B, see also Tables S3, S4), MA (Fig. 6B), retrolateral view of Eb (Fig. 4C), and Fu in both views (Fig. 7) cannot fully separate “hydatostella” from “rotundus”.

Figure 3 Shape component projections of retrolateral view of the right lateral subterminal apophysis after palp expansion on the first two principal component (PC) axes.

(A) “Kalanoro” group. (B) “Hydatostella” group. The symbols outside of the axes show how shapes change along each PC axis. Gray lines: the shape consensus at the maximum (SM) or minimum (Sm) of the PC axes; grey points: landmarks of the mean shape consensus of all specimens; black arrows: vectors showing landmarks movement from the mean shape consensus to SM or Sm.

Figure 4 Shape component projections of the male embolus on the first two principal component (PC) axes.

(A and B) Ventral view of the left embolus: (A) “Kalanoro” group; (B) “Hydatostella” group. (C and D) Retrolateral view of the right embolus after palp expansion: (C) “Kalanoro” group; (D) “Hydatostella” group. The symbols outside of the axes show how shapes change along each PC axis. Gray lines: the shape consensus at the maximum (SM) or minimum (Sm) of the PC axes; grey points: landmarks of the mean shape consensus of all specimens; black arrows: vectors showing landmarks movement from the mean shape consensus to SM or Sm.

Figure 5 Size comparisons of the selected male genital characters among the five morphospecies.

(A) Diameter of embolic ring. (B) Relative length of cymbium. Bold line: median; upper margin of the box: first quartile (Q1); lower margin of the box: third quartile (Q3); upper whisker: the maximum or Q1 + 1.5 × interquartile range (IQR); lower whisker: the minimum or Q3 − 1.5 × IQR; hollow circle: outlier; n: number of specimen(s).

Figure 6 Shape component projections of ventral view of the left median apophysis on the first two principal component (PC) axes.

(A) “Kalanoro” group. (B) “Hydatostella” group. The symbols outside of the axes show how shapes change along each PC axis. Gray lines: the shape consensus at the maximum (SM) or minimum (Sm) of the PC axes; grey points: landmarks of the mean shape consensus of all specimens; black arrows: vectors showing landmarks movement from the mean shape consensus to SM or Sm.

Figure 7 Shape component projections of the male fulcrum on the first two principal component (PC) axes.

(A and B) Ventral view of the left fulcrum: (A) “Kalanoro” group; (B) “Hydatostella” group. (C and D) Retrolateral view of the right fulcrum after palp expansion: (C) “Kalanoro” group; (D) “Hydatostella” group. The symbols outside of the axes show how shapes change along each PC axis. Gray lines: the shape consensus at the maximum (SM) or minimum (Sm) of the PC axes; grey points: landmarks of the mean shape consensus of all specimens; black arrows: vectors showing landmarks movement from the mean shape consensus to SM or Sm.

Figure 8 Shape component projections of the posterolateral view of the left retroalteral tibial apophysis on the first two principal component (PC) axes.

(A) “Kalanoro” group. (B) “Hydatostella” group. The symbols outside of the axes show how shapes change along each PC axis. Gray lines: the shape consensus at the maximum (SM) or minimum (Sm) of the PC axes; grey points: landmarks of the mean shape consensus of all specimens; black arrows: vectors showing landmarks movement from the mean shape consensus to SM or Sm.

Female genital characters

The shape of MF can well separate all morphospecies in both groups (Fig. 9). The epigynal margin (Fig. 10) and the vulva (Fig. 11) can only support partial or no separation among morphospecies within their group. Within the “Kalanoro” group, all three analyses of the female genital structures agree that D. kalanoro and “bedjanic” are different (Figs. 9A–11A). The morphospecies “gregoric”, however, has an epigynal margin in the shape between those of the other two morphospecies (Fig. 10A) and a vulva similar to D. kalanoro (Fig. 11A). All three morphospecies are different in the shape of MF (Fig. 9A), mainly in the presence or absence of the horn extension at the MF (PC1), the aspect ratio of the MF (PC1 & 2), and the length of the EF (PC2). Within the “Hydatostella” group, the two morphospecies are well separated by the epigynal margin (Fig. 10B) and the MF (Fig. 9B). The morphospecies “hydatostella” has a pentagon-shaped epigynal margin (Fig. 10B, PC1) with shorter EF (Fig. 9B, PC1) while “rotundus” has a round or triangular epigynal margin with relatively longer EF.

Figure 9 Shape component projections of ventral view of the epigynal middle field on the first two principal component (PC) axes.

(A) “Kalanoro” group. (B) “Hydatostella” group. The symbols outside of the axes show how shapes change along each PC axis. Gray lines: the shape consensus at the maximum (SM) or minimum (Sm) of the PC axes; grey points: landmarks of the mean shape consensus of all specimens; black arrows: vectors showing landmarks movement from the mean shape consensus to SM or Sm.

Figure 10 Shape component projections of ventral view of the epigynal margin on the first two principal component (PC) axes.

(A) “Kalanoro” group. (B) “Hydatostella” group. The symbols outside of the axes show how shapes change along each PC axis. Gray lines: the shape consensus at the maximum (SM) or minimum (Sm) of the PC axes; grey points: landmarks of the mean shape consensus of all specimens; black arrows: vectors showing landmarks movement from the mean shape consensus to SM or Sm.

Figure 11 Shape component projections of dorsal view of the vulva arrangement on the first two principal component (PC) axes.

(A) “Kalanoro” group. (B) “Hydatostella” group. The symbols outside of the axes show how shapes change along each PC axis. Gray lines: the shape consensus at the maximum (SM) or minimum (Sm) of the PC axes; grey points: landmarks of the mean shape consensus of all specimens; black arrows: vectors showing landmarks movement from the mean shape consensus to SM or Sm. Note: The vulva of a female “bedjanic” (KPARA00144) was excluded from this analysis due to severe deformation caused by a Mantispidae larva parasitizing her epigastric furrow beneath the epygynal middle field (see Fig. S5).

Morphometric summary

The first two PC axes in all the analyses explain more than half of the shape variations (Table S5). Separation of the five hypothesized morphospecies is supported by different combinations of the characteristics (Figs. 12–14). Measurement of the somatic characteristics in general cannot support the separation of morphospecies but can support the two species groups. Only shape differences of the three structures out of ten, LA, Eb (ventral view), and MF, can separate all the morphospecies in both groups (Figs. 12–14). Degrees of variation in the shape of the other seven structures and the measurements of male genitalia differ between groups; these features can only provide evidence for some, but not all, morphospecies (Figs. 2–11). The shapes of the Fu, in both views, show higher variation compared to that of the other structures and fail to separate most morphospecies (Fig. 7). However, the shape of Fu can potentially help to diagnose the major clades across Dolomedes phylogeny (see Taxonomy).

Figure 12 Shape consensuses of the retrolateral view of the embolus, fulcrum, and lateral subterminal apophysis of the expanded right male palp in each morphospecies.

Pairs of shapes covered in grey squares represent those that cannot separate morphospecies based on the two principle component axes. The three morphospecies left to the black line belong to the “Kalanoro” group and the two right to the black line are the “Hydatostella” group.

Figure 13 Shape consensuses of the ventral view of the median apophysis, embolus, fulcrum, and the posterolateral view of the retrolateral tibial apophysis of the left male palp in each morphospecies.

Pairs of shapes covered in grey squares represent those that cannot separate morphospecies based on the two principle component axes. The three morphospecies left to the black line belong to the “Kalanoro” group and the two right to the black line are the “Hydatostella” group.

Figure 14 Shape consensuses of the ventral view of the margin of the epigynum, epigynal middle field, and the dorsal view of the vulva of the females in each morphospecies.

Pairs of shapes covered in grey squares represent those that cannot separate morphospecies based on the two principle component axes. The three morphospecies left to the black line belong to the “Kalanoro” group and the two right to the black line are the “Hydatostella” group.

Phylogenetic and species delimitation analyses

We amplified 46 COI sequences with 1,192 base pairs from eight D. kalanoro, five “gregoric”, 16 “bedjanic”, four “hydatostella”, five “rotundus”, and eight juveniles of the “Kalanoro” group. Both ML and BI analyses recover monophyly of our morphospecies, and these are well supported at least in one, if not both, analyses (Fig. 15). The two species groups, “Kalanoro” and “Hydatostella”, are also well-supported as sister clades (Fig. 15). Within the “Kalanoro” group, the morphospecies “bedjanic” is sister to the clade uniting D. kalanoro and “gregoric” (Fig. 15). Genetic distances among these morphospecies hover between 3 and 5% (Fig. 15). Analyses show a weak population structure within “bedjanic” with specimens from Marojejy differing from those from Analamazoatra + Andasibe-Mantadia by less than 2% (Fig. 15). Genetic distances between morphospecies of the “Hydatostella” are closer to 3% (Fig. 15). Both ASAP and bPTP support the five morphospecies (Fig. 15).

Figure 15 Summarized COI gene tree and species delimitations with nodes strongly supported by maximum likelihood and Bayesian inference analyses marked by solid circle and square, respectively.

The color-coded bars represent genetic distances between major clades and the separation of the species under different species delimitation methods; letters at each branch represent species clades: (A) Dolomedes hydatostella sp. nov. (B) Dolomedes rotundus sp. nov. (C) Dolomedes bedjanic sp. nov. (D) Dolomedes gregoric sp. nov. (E) Dolomedes kalanoro Silva & Griswold, 2013; ASAP: Assemble Species by Automatic Partitioning (Puillandre, Brouillet & Achaz, 2021; bPTP: Bayesian implementation of the Poisson tree process model analysis (Zhang et al., 2013); ML: maximum likelihood; PP: posterior probability. Images (A–E) show the habitus of each species.

Habitat preferences and geological barriers

We observed coexistence of several Dolomedes species in the same river drainage and even in the same river sections or water bodies (Fig. 16, Table 2). Namely, “bedjanic” and “hydatostella” shared waterbodies at Marojejy (Figs. 16C and 16D). The three morphospecies of the “Kalanoro” group and “rotundus” were collected in the connected river system (Fig. 16, see also Fig. 1 in Kramer et al., 1997) in Andasibe-Mantadia and Analamazoatra. Their coexistence in highly connected river and water systems would estimate low levels of geological barriers. However, as explained below, even the species that coexist occupy different types of habitats (Fig. 17).

Figure 16 The collecting localities of the five Dolomedes species described in the study.

Table 2 Habitat classification for each locality in Madagascar where Dolomedes specimens were collected and recorded; see Taxonomy for coordinates.

Locality	Species recorded	Habitat classification	
Canopy	Velocity	
The 2nd muddy stream and swamp below Mantella Camp, Marojejy	hydatostella	Dense	Standing	
The slow flowing stream and swamp along trail Kalanoro, Vakona Lodge, Andasibe-Mantadia	bedjanic, rotundus	Dense	Standing	
Slow flowing part of the streams around Lac Vert, Analamazoatra	rotundus	Dense	Standing	
The river next to Hotel Feon’ ny Ala, Analamazoatra	kalanoro	Open	Flowing	
The river along trail Circuit Tsakoka, Andasibe-Mantadia	kalanoro	Open	Flowing	
The 1st, 3rd–5th stream below Mantella Camp, Marojejy	bedjanic	Dense	Flowing	
The stream on the trail toward Cascade de Humbert, Marojejy	bedjanic	Dense	Flowing	
The streams around Lac Vert, Analamazoatra	kalanoro, bedjanic	Dense	Flowing	
The stream along trail Chute Sacree, Andasibe-Mantadia	gregoric, bedjanic	Dense	Flowing	

Figure 17 Estimated habitat preferences of the five Dolomedes species as summarized from Table 2.

Estimations of the habitat preferences suggest the morphospecies of the “Kalanoro” group occupy different types of habitats with partial overlaps; while “hydatostella” and “rotundus” inhabit similar habitats (Table 2, Fig. 17). The three morphospecies of the “Kalanoro” group coexist and can all occupy habitats with flowing water and dense canopy coverage, but in addition, D. kalanoro and “bedjanic” also occupy habitats with flowing water and open canopy, as well as standing water and dense canopy, respectively (Table 2, Fig. 17). The two morphospecies of the “Hydatostella” group show similar habitat preferences, both inhabiting habitats with standing water and dense canopy (Table 2, Fig. 17), albeit in two localities without connections of waterbodies.

Discussion

Our model demonstrates how an integrative approach can improve the accuracy in spider taxonomy. Dolomedes contains both extremely widespread as well as locally endemic species that often coexist in any biogeographic region (World Spider Catalog, 2023; see also Raven & Hebron, 2018). Therefore, regional reviews could result in conflicting species boundaries based on different combinations of morphological evidence (see Zhang, Zhu & Song, 2004; Tanikawa & Miyashita, 2008). By investigating variation in diagnostic characteristics under an integrative taxonomic model, our study, while limiting its focus on the Madagascar Dolomedes species, facilitates a comparative framework potentially applicable to any Dolomedes taxonomic research.

In our study, two groups of Dolomedes, the “Kalanoro” and the “Hydatostella” group, show continuous or minor differences in the genital structures among exemplars. It was therefore difficult to define whether these differences were intra- or interspecific. We found our delimitations, based on molecular and ecological, in addition to the morphological evidence, to be decisive in solving this taxonomic problem. Our integrative model can well separate Dolomedes kalanoro and the hypothesized four morphospecies collected from the humid forest of north and east Madagascar. The species boundaries receive support from all three types of evidence and could form preliminary hypotheses of their speciation, as we discuss below. We hence describe the four hypothesized morphospecies as new species, namely D. gregoric sp. nov., D. bedjanic sp. nov., D. hydatostella sp. nov., and D. rotuntus sp. nov. (see Taxonomy).

Our model strongly suggests that relying solely on a few characteristics to diagnose Dolomedes species can be risky and that instead multiple types of evidence should be considered when establishing species boundaries. Our results reveal that the combinations of diagnostic characteristics differ even among closely-related Dolomedes. In this study, only the female median field and the male lateral subterminal apophysis and embolus can diagnose species of both groups of Madagascar Dolomedes. Therefore, we recommend male palp expansion for examining the lateral subterminal apophysis, a structure rarely described in Dolomedes taxonomy. We note that palpal expansion is an irreversible manipulation and as such may not be appropriate for those Dolomedes species known only from limited type material or historical specimens. In such cases, we recommend our proposed integrative taxonomic approach to be authorized first by curators or collection managers.

Structures involved in copulation are under strong selection pressure (Kuntner, Coddington & Schneider, 2009; Kuntner et al., 2016) and should therefore show stability in size and shape within species. In this study, embolus and copulatory opening (shaped by the epigynal folds and median field) are such structures related to sperm transfer, and as such maintain relatively low intraspecific variation. They can reliably diagnose species in both groups of Madagascar Dolomedes. In some taxa, RTA is responsible for anchoring the male palp into the correct position during copulation (Gering, 1953; Jäger, 2020; Poy et al., 2023) and is considered a critical diagnostic characteristic. However, RTA in Dolomedes sometimes presents relatively high degrees of variation (Tanikawa & Miyashita, 2008; Vink & Dupérré, 2010; see also Silva, Gibbons & Sierwald, 2015). While the RTA in Dolomedes tenebrosus Hentz, 1844 facilitates locking the distal sclerotized tube into copulation position (Sierwald & Coddington, 1988), where and how RTA anchors on the female genitalia in Dolomedes remains unknown. Indeed, our knowledge of how each genital structure functions and interacts during copulation in Dolomedes is very limited. More basic research is needed to further study the genital variation and its utility in species diagnosis.

A combination of habitat preferences, geological barriers, and dispersal-related traits can facilitate preliminary hypotheses of speciation among the currently five known Madagascar Dolomedes species. Based on the relative length of leg I and tarsus I, morphospecies of the “Kalanoro” group are good dispersers with better walking abilities and aquatic locomotion compared to the “Hydatostella” group. They also prefer flowing water which compared with standing water provides better chances of dispersal. Therefore, the “Kalanoro” species might be able to maintain gene flow across longer geological barriers among unconnected river drainages. This generalization already holds in Dolomedes bedjanic sp. nov., which we collected in disjunct parts of north and east Madagascar. Despite having good dispersal abilities and higher chances of dispersal, adaptation to different habitats may support the gene flow limitation among the three morphospecies of the “Kalanoro” group. On the other hand, the “Hydatostella” group is a relatively poor disperser and dwells in standing waterbodies under a dense canopy. Therefore, isolation may easily take place between populations whose water bodies are disjunct. Such isolation may promote diversification, and thus we expect more species of the “Hydatostella” group to be discovered in Madagascar.

Our discoveries in Madagascar Dolomedes, albeit incomplete, include both wide ranging species like D. kalanoro and D. bedjanic sp. nov., as well as local endemics, D. gregoric sp. nov., D. hydatostella sp. nov., and D. rotundus sp. nov. Such pattern is known in many other organisms on Madagascar, such as lemurs (Wilmé, Goodman & Ganzhorn, 2006), geckos (Pearson & Raxworthy, 2009), as well as in other spiders (Agnarsson & Kuntner, 2005; Agnarsson et al., 2015; Jäger, 2020; Griswold et al., 2022). The two major biogeographic scenarios behind the high local endemism rates are the climatic gradient hypothesis (Smith et al., 1997; Schneider et al., 1999) and the watershed hypothesis (Wilmé, Goodman & Ganzhorn, 2006). The former considers the complex climate gradient across the island to shape high levels of local endemism (see also Antonelli et al., 2022). The second hypothesis suggests that the isolated humid refuges formed by rivers that originated from mountains during Quaternary climate shifts have promoted local endemism (Wilmé, Goodman & Ganzhorn, 2006). These hypotheses are not mutually exclusive, but could in combination explain the observed contemporary distribution patterns of much of the biodiversity on Madagascar (Pearson & Raxworthy, 2009; Brown et al., 2014). Considering that freshwater-associated organisms are sensitive to spatial pattern changes in water bodies and river drainages, be it natural (e.g., Griffiths, 2006; Huang & Lin, 2011; Dias et al., 2013; Ye et al., 2018; Chen et al., 2021) or artificial (Raharimalala et al., 2012), the two hypotheses could well explain the biogeography and speciation of Madagascar Dolomedes.

Our research largely increases our knowledge of the semi-aquatic fauna in Madagascar. Despite recent advances in pisaurid taxonomy in Madagascar (Silva & Griswold, 2013a, 2013b; Silva & Sierwald, 2013), only a single species of Dolomedes has been deemed valid prior to our study. Our addition of four new species on the island—a result from only surveying two national parks—only points to how incomplete the taxonomy of Dolomedes in Madagascar really is. Their rarity in collections is likely due to nocturnal and semi-aquatic lifestyles. Indeed, the species described here strictly adhere to water bodies and are never found on trails. In addition, these Dolomedes species are cryptic and inactive during the day, but can be more easily located at night. Consequently, arachnologists routinely overlook Dolomedes species if they do not engage in targeted sampling along water bodies at night. Considering all above, we expect that future targeted work in yet unexplored parts of this vast island will uncover many more Dolomedes species. This stresses the importance of protecting aquatic habitats, however small, in Madagascar and elsewhere. Considering their poor dispersal abilities and narrow habitat preferences, the locally endemic Dolomedes, especially the two “Hydatostella” species, could rapidly go extinct in the face of small water body degradation.

Taxonomy

Family Pisauridae Simon, 1890

Dolomedes Latreille, 1804

Cispiolus Roewer, 1955. Type species: C. upembensis Roewer, 1955. Synonymized by Blandin (1979).

Teippus Chamberlin, 1924. Type species: T. lamprus Chamberlin, 1924. Synonymized by Carico (1973), after Gertsch (1934).

Type species: Araneus fimbriatus Clerck, 1757: 106; plate 5, fig 9.

Clerck’s (1757) description of a male “Araneus fimbriatus” from Sweden is the first description of any Dolomedes species. When Latreille (1804) established the genus Dolomedes, Araneus fimbriatus became its type species.

Diagnosis: Dolomedes can be separated from all the other pisaurid genera except Mangromedes Raven & Hebron, 2018, Megadolomedes Davies & Raven, 1980, Tasmomedes Raven & Hebron, 2018, Bradystichus Simon, 1884, Caledomedes Raven & Hebron, 2018, and Ornodolomedes Raven & Hebron, 2018, by the combination of the following characters:

Male palp (see also Sierwald, 1990; Santos, 2007) 1) The presence of Sa, a round shaped and sclerotized part of upper T. This structure is considered as a reduced DTA in Santos (2007) cladistic analyses (Fig. 18, red arrow).

2) The presence of LA, which originated from the DST together with the Eb and Fu (Fig. 19).

Figure 18 Comparisons of the male palps among the five Dolomedes species.

(A and F) Dolomedes kalanoro Silva & Griswold, 2013 (KPARA00185). (B and G) D. gregoric sp. nov. (Holotype male, USNMENT01580825). (C and H) D. bedjanic sp. nov. (Holotype male, USNMENT01580827). (D and I) D. hydatostella sp. nov. (Holotype male, USNMENT01580829). (E and J) D. rotundus sp. nov. (Holotype male, USNMENT01580831). (A–E) Ventral view. (F–J) Retrolateral view. Red arrows point to the saddle; black arrows show the lateral lobe on the fulcrum.

Figure 19 Retrolateral view of the right distal sclerotized tube of males of the five Dolomedes species.

(A) Dolomedes kalanoro Silva & Griswold, 2013 (KPARA00185). (B) D. gregoric sp. nov. (Holotype male, USNMENT01580825). (C) D. bedjanic sp. nov. (Holotype male, USNMENT01580827). (D) D. hydatostella sp. nov. (Holotype male, USNMENT01580829). (E and F) D. rotundus sp. nov.: (E) Holotype male (USNMENT01580831); (F) non type specimen (RMCA, MT_207084), showing the shape variation. The black arrow shows the dorsal direction. Eb, embolus; Fu, fulcrum; LA, lateral subterminal apophysis.

Female epigynum (see also Sierwald, 1989) 1) Small knob, bulb, or horn shaped HS, which is also called AB (Figs. 20A–20E, red arrows). HS in other pisaurids is usually extended and elongated.

2) Relatively long and usually vertically coiled FD that starts with a tubular part and ends with a flat flake (Figs. 20A–20E).

Figure 20 Comparisons of the female epigynum of the five Dolomedes species.

(A and F) Dolomedes kalanoro Silva & Griswold, 2013 (KPARA00193). (B and G) D. gregoric sp. nov. (Paratype female, USNMENT01580826). (C and H) D. bedjanic sp. nov. (Paratype female, USNMENT01580828). (D and I) D. hydatostella sp. nov. (Paratype female, USNMENT01580830). (E and J) D. rotundus sp. nov. (Paratype female, USNMENT01580832). (A–E) dorsal view with a line drawing showing the arrangement of the vulva, the circle represents the end of copulatory duct. (F– J) Ventral view, red arrows show the accessory bulb.

Dolomedes can be distinguished from Mangromedes by i) the straight and non pseudo-segmented leg tarsi (but, see Santos, 2007); ii) the female COp positioned at the middle or the posterior part of epigynum and anteriorly or laterally opened (Figs. 20F–20J); iii) the Eb and Fu that are positioned at the retrolateral side of the palp and near vertically curved or coiled (Figs. 18A–18E); and iv) the male palp RTA that does not deeply divide into two branches (Figs. 18F–18I and 21, see also Raven & Hebron, 2018).

Figure 21 Posterolateral view of the left retrolateral tibial apophysis of the five Dolomedes species.

(A) Dolomedes kalanoro Silva & Griswold, 2013 (KPARA00185). (B) D. gregoric sp. nov. (Holotype male, USNMENT01580825). (C) D. bedjanic sp. nov. (Holotype male, USNMENT01580827). (D) D. hydatostella sp. nov. (Holotype male, USNMENT01580829). (E) D. rotundus sp. nov. (Holotype male, USNMENT01580831).

Dolomedes can be distinguished from Megadolomedes by i) the straight and non pseudo-segmented leg tarsi (but, see Santos, 2007); ii) the simply curved or coiled Eb and Fu with less than one loop (Fig. 19, but see D. bistylus Roewer, 1955) and the embolic ring not extending anteriorly to near the tip of the Cy (Figs. 18A–18E); and iii) the broad female CD (Figs. 20A–20E) without broken male embolus as mating plug (Davies & Raven, 1980; Raven & Hebron, 2018).

Dolomedes can be distinguished from Tasmomedes by i) the simply curved or coiled Eb and Fu (Fig. 19) that do not extend anteriorly to near the tip of the Cy (Figs. 18A–18E); ii) the strong and fully developed male palp RTA (Fig. 21); and iii) vertically coiled female FD (Figs. 20A–20E).

Dolomedes can be distinguished from Bradystichus by i) the unmodified abdomen (Fig. 22); ii) the absence of high-contrast ventral abdominal coloration (see Raven & Hebron, 2018); iii) the denser leg femur spines, five pairs in Dolomedes while only three in Bradystichus (see Raven & Hebron, 2018); iv) the fully developed unpaired tarsal claw (Platnick & Forster, 1993; Raven & Hebron, 2018), and v) the absence of prodorsal scopula on male Cy (Fig. 18) (Platnick & Forster, 1993; Raven & Hebron, 2018).

Figure 22 Dolomedes species collected from humid forests in the east and the north of Madagascar, showing the habitus coloration and variation.

(A and B) Dark morph D. kalanoro Silva & Griswold, 2013: (A) A female (KPARA00184) on a rock in a river; (B) a male (KPARA00185) at a river bank. (C and D) Dark morph D. gregoric sp. nov.: (C) A female (KPARA00250) in hunting pose on water; (D) a male (KPARA00248) placed on a white background. (E–F): D. bedjanic sp. nov.: (E) A female (KPARA00129) on a rock in a stream; (F) a male (KPARA00234) on shallow water under vegetation. (G and H) White banded morph D. kalanoro: (G) A male (KAPAR00227) hiding in a dead tree above a river during day time; (H) a female carrying an egg sac hiding in a tree trunk near a river during day time. (I) White banded morph D. gregoric sp. nov. (Holotype male, USNMENT01580825) on a tree trunk near a river. (J and K) D. hydatostella sp. nov.: (J) A female (KPARA00163) in a shallow understory swamp; (K) a male (KPARA00258) placed on a white background. (L and M) D. rotundus sp. nov.: (L) A female (KPARA00243), and (M) a male (KPARA00236) in a shallow part of a stream.

Dolomedes can be distinguished from Caledomedes by i) the AER that is wider than the posterior width of MOA; and ii) the shorter male palp RTA (Raven & Hebron, 2018).

Dolomedes can be distinguished from Ornodolomedes by i) the relatively simple carapace color pattern (Fig. 22, see also Raven & Hebron, 2018); and ii) the shorter leg tarsal and metatarsal spines that do not overlap (see Raven & Hebron, 2018).

Remarks. Subfamilies of Pisauridae are still debated (Simon, 1898; Petrunkevitch, 1928; Sierwald, 1990; Griswold, 1993; Santos, 2007; Murphy & Roberts, 2015; Polotow, Carmichael & Griswold, 2015; Silva, Gibbons & Sierwald, 2015; Albo et al., 2017; Wheeler et al., 2017; Piacentini & Ramírez, 2019) and have not been comprehensively analyzed under phylogenetic framework. Recent phylogenies suggest that Dolomedes and the New Caledonian endemic Bradystichus form a clade that is sister to all other pisaurids (Wheeler et al., 2017; Piacentini & Ramírez, 2019). Following the literature and the similarities in their genital morphology (see genus diagnosis), perhaps Dolomedes and its close relatives can be argued to form a subfamily (Dolomedinae). However, a well sampled, robust phylogeny and a taxonomic review at a global scale are needed to detect phylogenetic proximities and synapomorphies of pisaurid subfamilies.

Key to known Dolomedes species of Madagascar

1) Large and long-legged Dolomedes, female carapace width above 7 mm, and male carapace width above 6.5 mm (Fig. 2). Habitus uniformly brown to dark brown with light-colored, but usually indistinct, margins (Figs. 22A–22F). A few individuals have distinct but thin white lateral bands that do not expand to the edge of the carapace (Figs. 22G–22I): “Kalanoro” group

Female…………………………………………………………………………………….2

Male………………………………………………………………………………………4

Small and short-legged Dolomedes, female carapace width below 7 mm, and male carapace width below 6.5 mm (Fig. 2). Habitus dark brown to black with distinctively broad white lateral bands that expand to the edge of the carapace (Figs. 22I–22M): “Hydatostella” group

Female…………………………………………………………………………………….6

Male………………………………………………………………………………………7

2) Epigynum MF with a horn extension (Figs. 9A, 14, 20F): D. kalanoro Silva & Griswold, 2013

Epigynum MF without a horn extension (Figs. 9A, 14, 20G and 20H) ………………3

3) Epigynum MF long and narrow; CD long and connected to the BS ventrally; AB laterally positioned (Figs. 9A, 14, 20B): D. gregoric sp. nov.

Epigynum MF short and wide; CD short and connected to the BS anteriorly; AB anteriorly positioned (Figs. 9A, 14, 20C): D. bedjanic sp. nov.

4) Palp long, over twice the carapace width (Fig. 2H); Cy shorter than the palp tibia (Fig. 5B); RTA dorso-laterally positioned (Fig. 18G) without a dorsal lobe (Figs. 8A, 13, 21C); MA with a broad base (Figs. 6A, 13, 18C); Eb short (Figs. 4A–4B, 12, 19C), De below 1.0 (Fig. 5A); LA with a broad tip (Figs. 3A, 12, 19C): D. bedjanic sp. nov.

Palp shorter than twice the carapace width (Fig. 2H); Cy longer than the palp tibia (Fig. 5B); RTA laterally positioned with a dorsal lobe (Figs. 8A, 13, 21A and 21B); MA with a narrow base (Figs. 6A, 13, 18A and 18B); De above 1.0 (Fig. 5A); LA with a narrow tip (Figs. 3A, 12, 19A and 19B) …………………………………………………5

5) Embolus long with a narrow base (Figs. 4A and 4B, 12, 19A), De above 1.2 (Fig. 5A); Cy large, at least 1.2 times longer than the palp tibia (Fig. 5B); MA expands gradually from the base to the middle part (Figs. 6A, 13A, 18A); LA with a straight section at the base of the dorsal edge (Figs. 3A, 12, 19A): D. kalanoro Silva & Griswold, 2013

Embolus long with a broadened base (Figs. 4A–4B, 12, 19B), De between 1.0 and 1.2 (Fig. 5B); MA distinctly expands at the middle part (Figs. 6A, 13, 18B); LA without a straight section at the base of the dorsal edge (Figs. 3A, 12, 19B): D. gregoric sp. nov.

6) Epigynum pentagon-shaped with short EFs, posterior MF relatively long (Figs. 9B, 14, 20I): D. hydatostella sp. nov.

Epigynum round or triangular shaped with long EFs that extend to near the anterior edge of the epigynum; posterior MF relatively short (Figs. 9B, 14, 20J): D. rotundus sp. nov.

7) Dorsal and ventral lobes of RTA similar sized (Figs. 8B, 13, 21D); In the ventral view, retrolateral arc of the Eb straight (Figs. 4B, 13, 18D); LA with a distinct narrow base (Figs. 3B, 12, 19D): D. hydatostella sp. nov.

Dorsal lobe of RTA round and broadened, larger than the ventral lobe (Figs. 8B, 13, 21E); In the ventral view, retrolateral arc of the Eb bent (Figs. 4B, 13, 18E); LA without a distinct narrow base (Figs. 3B, 12, 19E and 19F): D. rotundus sp. nov.

Species group “Kalanoro”

Diagnosis. Dolomedes species of the “Kalanoro” species group can be distinguished from all the other known Dolomedes, except the Madagascar Dolomedes species of the group “Hydatostella”, by the combination of the following characters:

Male: The presence of a lateral lobe at the retrolateral edge of the Fu in the ventral view (Fig. 18, black arrows; see species descriptions for the structure in D. rotundus).

Female: i) epigynum longer than wide, without lateral extensions at the posterior edge (Figs. 20F–20J); ii) two fully separated, medially positioned, relatively small, and triangular or square-shaped MF windows (Figs. 20F–20J); iii) CD wider than the first loop of the FD, or in similar width (Figs. 20A–20E); and iv) FD vertically coiled without any contrary flexure or horizontal spiral (Figs. 20A–20E).

Dolomedes of the group “Kalanoro” can be distinguished from those of the “Hydatostella” group by i) larger body sizes, ii) longer appendages, and iii) brownish coloration without distinct white lateral bands; the lateral bands of the white banded morph do not extend to the edge of the carapace.

Composition. Dolomedes kalanoro, D. gregoric sp. nov., D. bedjanic sp. nov.

Dolomedes kalanoro Silva & Griswold, 2013

(Figs. 16A, 18A, 18F, 19A, 20A, 20D, 21A, 22A, 22B, 22G, 22H, 23, 24)

Figure 23 Male Dolomedes kalanoro Silva & Griswold, 2013 (KPARA00185).

(A) Dorsal view. (B) Anterior view. (C and D) Left palp: (C) Ventral view; (D) retrolateral view showing the length ratio between cymbium and tibia. (E and F) Left palp: (E) Ventral view; (F) retrolateral view showing anatomical details. BCA: basal cymbium apophysis; Co: conductor; Cy: cymbium; DTP: distal tegular projection; Eb: embolus; Fu: fulcrum; MA: median apophysis; RTA: retrolateral tibial apophysis; Sa: saddle; ST: subtegulum; T: tegulum; VTA: ventral tibial apophysis.

Figure 24 Female Dolomedes kalanoro Silva & Griswold, 2013 (KPARA00193).

(A) Dorsal view. (B) Anterior view. (C and E) Epigynum: (C) ventral view; (D) lateral view, the arrow showing the horn extension; (E) dorsal view. CD: copulatory duct; COp: copulatory opening; EF: epigynal fold; FD: fertilization duct; LL: lateral lobe; MF: middle field.

Dolomedes kalanoro Silva & Griswold, 2013: 462; fig 2–12 (Description of male).

Material examined. MADAGASCAR: Toamasina Province: two females, two males, and two juveniles, the river next to Hotel Feon’ny Ala (18°56′49.9″S, 48°25′8.8″E, 942 m), 4 IV 2022, Kuang-Ping Yu (KPY) leg., KPARA00183–187, 00201 (NIB); one female, two males, and four juveniles, the streams around Lac Vert, Reserve Analamazoatra (18°56′14.2″S, 48°25′12.5″E, 939 m), 4–8 IV 2022, KPY leg., KPARA00193–196, 00212–213, 00228, 00231 (NIB); one female, one male, and one juvenile, the river along Circuit Tsakoka, Parc national d’Andasibe-Mantadia (18°47′54.5″S, 48°25′34.8″E, 959 m), 6 IV 2022, KPY leg., KPARA00207–208, 00214 (NIB).

Other material examined. Dolomedes straeleni Roewer, 1955: Holotype female, P.N. Upemba, Congo (1,750 m), 14–31 III 1947, Byue-Bala, affl. g. Muye et sous-affl. dr. Lufira, MT_119613 (RMCA); Paratype female, ParcNal. Upemba, Congo, SMF_ RII/10547 (SMF). See Remark for species validity.

Diagnosis. Male D. kalanoro differs from the other two Dolomedes species of the “Kalanoro” group by i) the MA expands gradually from the narrow base (Figs. 18A, 23C and 23E); ii) the long and narrow-based Eb (Figs. 18A, 19A, 23C and 23E); and iii) the LA with a narrow ventral tip and a straight section at the basal part of the dorsal edge (Fig. 19A). Female D. kalanoro can be separated from all the other Dolomedes species by the narrow, horn shaped, and anteriorly protruded extension on the ventral epigynal MF (Figs. 20D, 24C and 24E; but see Remark).

Description. Male (KPARA00185). Total length 15.18: carapace length 7.89, width 7.11, anterior height 2.78, posterior height 3.42; abdomen length 7.29, width 3.33. Length of palp and legs: palp 13.81 (5.64, 2.21, 2.61, 3.35); leg I 41.89 (10.81, 3.35, 11.16, 10.47, 6.10); leg II 41.47 (11.01, 4.13, 10.41, 10.61, 5.31); leg III 31.58 (10.02, 3.79, 8.87, 8.90, NA); leg IV 43.33 (11.10, 4.13, 10.67, 11.40, 6.03). Leg formula 4123. Carapace pear-shaped, light brown with dense black short setae that form series of black radial markings pointing towards the distinct fovea (Figs. 22B, 23A). A pair of triangular black markings positioned anterior to the fovea (Figs. 22B, 23A). The margin of the carapace covered with sparse milky white short setae that form indistinct lateral bands (Figs. 22B, 23A). Eight eyes ringed with black (Fig. 23B). Eyes arranged in two rows. AER weakly recurved while PER strongly recurved. MOA dark brown with several black setae (Fig. 23B). Diameters of AME 0.34, ALE 0.24, PME 0.48, PLE 0.54; MOA length 1.04, anterior width of MOA 0.74, posterior width of MOA 1.23; interval of AMEs 0.10, interval of PMEs 0.19, interval between AME and ALE 0.13, interval between PME and PLE 0.16. Clypeus 1.01, brownish covered with dark short setae. Chelicera length 3.18, chestnut-brown covered with black long setae (Fig. 23B). Both chelicerae with three promarginal and four retromarginal teeth, both fangs and marginal teeth black. Endite length 2.52, width 1.23; labium length 1.43, width 1.45; sternum near round, length 3.10, width 3.38; all endites, labium, and sternum yellowish-brown. Abdomen long oval with a distinct brownish cardiac mark; dorsum with dense, dark brown short setae (Figs. 22B, 23A). Lateral abdomen covered with grayish short setae that form irregular and indistinct lateral bands. A series of white lines and spots distributed on the dorsal abdomen (Figs. 22B, 23A). Venter abdomen brown. Legs yellowish-brown, covered with dark blackish setae in different densities, regions with sparser setae form light-colored linear markings. Palp tibia with a highly sclerotized RTA divided into a sharp, larger ventral lobe and a smaller blunt dorsal lobe (Fig 21A). Basal retrolateral edge of the ventral Cy with an oval BCA (Figs. 18A, 23C and 23E). T sclerotized with a membranous upper edge, the prolateral side attached to the DTP and the retrolateral side attached to the membranous Co (Figs. 18A, 23C and 23E). T + DTP + Co forms a “U” shaped tegular ring (Figs. 18A, 23C and 23E). A highly sclerotized Sa sits at the lower center of the ring, attached to the T (Figs. 18A, 23C and 23E). MA hook-shaped, gradually expands from the narrow base and sits retrolaterally to the Sa (Figs. 18A, 23C and 23E). Fu hook shaped, retrolateral side with a broad and distinct lateral lobe (Figs. 18A, 23C and 23E). Ventral edge of the Fu folded and forms a groove that contains the long, narrow, and curved Eb (Figs. 18F, 19A, 23D and 23F). LA trapezoid with a blunt and narrow ventral tip (Fig. 19A). Dorsal edge of the LA with straight basal section (Fig. 19A). All Fu, Eb, and LA originated from the DST (Fig. 19A).

Female (KPARA00193). Total length 20.28: carapace length 10.31, width 9.32, anterior height 3.35, posterior height 4.17; abdomen length 9.97, width 5.78. Length of palp and legs: palp 14.51 (5.20, 2.12, 2.91, 4.28); leg I 43.23 (11.84, 4.99, 11.62, 9.45, 5.33); leg II 45.16 (12.73, 5.15, 11.65, 10.19, 5.44); leg III 40.05 (11.55, 4.41, 10.20, 9.17, 4.72); leg IV 46.26 (12.18, 4.88, 11.25, 11.79, 6.16). Leg formula 4213. Diameters of AME 0.44, ALE 0.29, PME 0.59, PLE 0.62; MOA length 1.32, anterior width of MOA 0.89, posterior width of MOA 1.45; interval of AMEs 0.16, interval of PMEs 0.25, interval between AME and ALE 0.20, interval between PME and PLE 0.57. Clypeus 1.56. Chelicera length 3.87. Both chelicerae with three promarginal and four retromarginal teeth. Endite length 3.03, width 1.50; labium length 1.59, width 1.64; sternum near round, length 4.31, width 4.40. Female similar to male, but larger, darker in the coloration, and with relatively shorter legs (Figs. 24A and 24B). Epigynum pentagon shaped, longer than wide and highly sclerotized; divided into two LLs by the narrow rectangular MF with two distinct membranous windows (Figs. 20D, 24C). Posterior half of the MF ventrally protruded with a narrow horn extension pointed anteriorly (Figs. 20D, 24C, 24D). CD long, coiled around one loop and connected to the BS ventrally (Figs. 20A, 24E). AB small, knob shaped, and laterally positioned (Figs. 20A, 24E). FD begins with a coiled and tubular part; ends with a triangular flake (Figs. 20A, 24E).

Variation. Given as variation of four females followed by variation of five males in parentheses. Total length 21.35 ± 1.06 (16.98 ± 1.17): carapace length 10.31 ± 0.93 (8.65 ± 0.64), width 9.15 ± 0.95 (7.85 ± 0.64), anterior height 3.39 ± 0.45 (2.83 ± 0.14), posterior height 4.23 ± 0.52 (3.75 ± 0.31); abdomen length 11.04 ± 0.81 (8.34 ± 0.74), width 6.92 ± 0.89 (4.73 ± 1.08). Palp 14.88 ± 1.55 (15.06 ± 0.79); leg I 44.03 ± 4.06 (46.21 ± 2.59); leg II 44.66 ± 3.85 (45.43 ± 2.31); leg III 40.44 ± 3.39 (40.00 ± 2.00); leg IV 48.74 ± 5.45 (46.65 ± 2.06). Diameters of AME 0.41 ± 0.06 (0.36 ± 0.03), ALE 0.29 ± 0.03 (0.25 ± 0.01), PME 0.56 ± 0.05 (0.51 ± 0.03), PLE 0.63 ± 0.02 (0.55 ± 0.02); Clypeus 1.48 ± 0.18 (1.10 ± 0.09). Chelicera length 3.96 ± 0.20 (3.34 ± 0.15). Endite length 3.06 ± 0.28 (2.58 ± 0.18), width 1.59 ± 0.19 (1.20 ± 0.08). Labium length 1.60 ± 0.18 (1.38 ± 0.08), width 1.88 ± 0.25 (1.52 ± 0.08). Sternum length 4.11 ± 0.40 (3.57 ± 0.31), width 4.34 ± 0.48 (3.73 ± 0.29). Dolomedes kalanoro has two different coloration morphs: the dark morph (Figs. 22A, 22B, 23A, 24A) has only light colored but indistinct carapace margin that fades in the anterior carapace and in the abdomen; the white banded morph (Figs. 22G, 22H) is more brownish with very distinct, but thin, white lateral bands that do not expand to the edge of the carapace. The former is more abundant in the surveyed region.

Natural history. Large sized wandering spider. Inhabits rivers with open canopy and streams with dense canopy (Fig 17, Table 2). Although spiders can be seen both day and night, they are more active at night. Spiders hide in cracks of river banks, gaps between roots of trees, and among dead tree trunks during day times (Figs. 22A–22H).

Remark. The male Dolomedes found in the eastern humid forest of Madagascar fits the original descriptions of D. kalanoro by Silva & Griswold (2013a) although the type specimens were collected in the western and the southern dry or subhumid forest of the island (Fig. 16A). We therefore consider these Dolomedes in Eastern Madagascar to be conspecific with D. kalanoro. The female D. kalanoro is very similar to D. straeleni collected and described from the Upemba Lake, Congo after re-examining the type series (K-P Yu, P Brogan, K Matjaž, 2023, unpublished data). Considering the lack of any male descriptions and molecular data available for detailed analyses and the historical biogeographic separation between Madagascar and Congo, our best hypothesis is that these are separate species.

Distribution. Western and southern dry and subhumid forests (Silva & Griswold, 2013a) and eastern humid forests of Madagascar (see Fig. 16A).

Dolomedes gregoric Yu & Kuntner sp. nov.

(Figs. 16B, 18B, 18G, 19B, 20B, 20E, 21B, 22C, 22D, 22I, 25, 26)

Figure 25 Male Dolomedes gregoric sp. nov. (Holotype, USNMENT01580825).

(A) Dorsal view. (B) Anterior view. (C and D) Left palp: (C) Ventral view; (D) retrolateral view showing the length ratio between cymbium and tibia. (E and F) Left palp: (E) Ventral view; (F) retrolateral view showing anatomical details.

Figure 26 Female Dolomedes gregoric sp. nov. (Paratype, USNMENT01580826).

(A) Dorsal view. (B) Anterior view. (C and D) Epigynum: (C) Ventral view; (D) dorsal view.

ZooBank id: urn:lsid:zoobank.org:act:075EEA19-43F1-4B65-97CC-5E014CE4D22C

Holotype male. MADAGASCAR: Toamasina Province: the stream along the trail Chute Sacree, Parc national d’Andasibe-Mantadia (18°49′30.7″S, 48°26′5.3″E, 976 m), 12 IV 2022, KPY leg., USNMENT01580825 (USNM).

Paratype female. Same collecting information as the Holotype, USNMENT01580826 (USNM).

Other material examined. Two females and one male, same collecting information as the Holotype, KPARA00248, 00250, 00254 (NIB).

Diagnosis. Male D. gregoric sp. nov. differs from the other two Dolomedes species of the “Kalanoro” group by i) the narrow-based MA that expands distinctly at the middle (Figs. 18B, 25C, 25E); ii) the long Eb with a broadened base (Figs. 18B, 19B, 25C, 25E); and iii) the LA with a narrow ventral tip but without a straight basal section at the dorsal edge (Fig. 19B). Female D. gregoric sp. nov. can be separated from the other two Dolomedes species of the “Kalanoro” group by i) the narrow MF without a horn extension (Figs. 20E, 26C); ii) the shorter EFs (Figs. 20E, 26C); and iii) the long CD coiled around a loop (Figs. 20B, 26D).

Description. Male (Holotype, USNMENT01580825). Total length 16.43: carapace length 8.27, width 7.39, anterior height 2.90, posterior height 3.80; abdomen length 8.16, width 4.65. Length of palp and legs: palp 14.84 (5.93, 2.40, 3.00, 3.51); leg I 47.24 (12.01, 4.55, 11.95, 11.56, 7.17); leg II 46.39 (12.16, 4.46, 11.81, 11.38, 6.58); leg III 40.71 (11.07, 3.99, 10.01, 10.15, 5.49); leg IV 47.74 (12.49, 4.25, 11.51, 12.62, 6.87). Leg formula 4123. Carapace pear-shaped, brownish with dense black short setae that form series of black radial markings pointing towards the distinct fovea (Fig. 25A). A pair of triangular black markings positioned anterior to the fovea (Fig. 25A). Lateral carapace with two distinct, thin white lateral bands formed by dense white setae that do not expand to the edge of the carapace (Fig. 25A). Eight eyes ringed with black (Fig. 25B). Eyes arranged in two rows. AER weakly recurved while PER strongly recurved. MOA dark brown with several black setae (Fig. 25B). Diameters of AME 0.41, ALE 0.30, PME 0.57, PLE 0.61; MOA length 1.21, anterior width of MOA 0.84, posterior width of MOA 1.32; interval of AMEs 0.13, interval of PMEs 0.21, interval between AME and ALE 0.12, interval between PME and PLE 0.45. Clypeus 1.04, brown covered with dark short setae (Fig. 25B). Chelicera length 3.18, chestnut-brown covered with black long setae (Fig. 25B). The right chelicera with four promarginal and five retromarginal teeth whereas the left chelicera with three and four teeth respectively. Both fangs and marginal teeth black. Endite length 2.61, width 1.19; labium length 1.17, width 1.56; sternum near round, length 3.63, width 3.84; all endites, labium, and sternum light brown. Abdomen long oval with a distinct brownish cardiac mark; dorsum with dense, dark brown short setae (Fig. 25A). Lateral abdomen covered with white short setae that form distinct lateral bands (Fig. 25A). Venter abdomen brown. Legs yellowish-brown, covered with dark blackish setae in different densities, regions with sparser setae form light-colored linear markings (Fig. 25A). Palp tibia with a highly sclerotized RTA divided into a sharp, larger ventral lobe and a blunt, smaller dorsal lobe (Fig. 21B). Basal retrolateral edge of the ventral Cy with an oval BCA (Figs. 18B, 25C and 25E). T sclerotized with a membranous upper edge, the prolateral side attached to the DTP and the retrolateral side attached to the membranous Co (Figs. 18B, 25C and 25E). T + DTP + Co forms a “U” shaped tegular ring (Figs. 18B, 25C and 25E). A highly sclerotized Sa sits at the lower center of the ring, attached to the T (Figs. 18B, 25C and 25E). MA sits retrolaterally to the Sa, hook-shaped with a narrow base and expands distinctly at the middle (Figs. 18B, 25C and 25E). Fu hook shaped, retrolateral side with a narrow and distinct lateral lobe (Figs. 18B, 25C and 25E). Ventral edge of the Fu folded and forms a groove that contains the long, broad-based, and curved Eb (Figs. 18G, 19B, 25D and 25F). LA trapezoid with a blunt and narrow tip (Fig. 19B). All Fu, Eb, and LA originated from the DST (Fig. 19B).

Female (Paratype, USNMENT01580826). Total length 17.02: carapace length 9.16, width 7.93, anterior height 3.16, posterior height 3.83; abdomen length 7.86, width 5.40. Length of palp and legs: palp 13.58 (4.67, 1.96, 2.81, 4.14); leg I 40.21 (10.67, 4.27, 10.85, 9.34, 5.08); leg II 41.35 (11.46, 4.53, 10.88, 9.38, 5.10); leg III 38.72 (10.90, 4.33, 9.53, 9.40, 4.56); leg IV 45.58 (11.70, 4.29, 11.38, 12.17, 6.04). Leg formula 4213. Diameters of AME 0.49, ALE 0.32, PME 0.65, PLE 0.63; MOA length 1.37, anterior width of MOA 1.02, posterior width of MOA 1.47; interval of AMEs 0.27, interval of PMEs 0.28, interval between AME and ALE 0.16, interval between PME and PLE 0.57. Clypeus 1.26. Chelicera length 3.24. Both chelicerae with three promarginal and four retromarginal teeth. Endite length 2.67, width 1.53; labium length 1.48, width 1.58; sternum near round, length 3.44, width 3.99. Female similar to male, but larger, darker in the coloration with indistinct light-colored carapace edge, and with relatively shorter legs (Figs. 26A, 26B). Epigynum pentagon shaped, longer than wide and highly sclerotized; divided into two LLs by the narrow rectangular MF with two small and distinct membranous windows (Figs. 20E, 26C). CD long, coiled around one loop and connected to the BS ventrally (Figs. 20B, 26D). AB small, knob-shaped, and laterally positioned (Figs. 20B, 26D). FD begins with a coiled and tubular part; ended with a spindle flake (Figs. 20B, 26D).

Variation. Given as variation of two females followed by variation of three males in parentheses. Total length 17.22 ± 0.58 (15.43 ± 1.42): carapace length 8.72 ± 0.42 (7.95 ± 0.46), width 7.68 ± 0.41 (7.15 ± 0.35), anterior height 3.04 ± 0.11 (2.77 ± 0.18), posterior height 3.58 ± 0.25 (3.64 ± 0.23); abdomen length 8.50 ± 0.67 (7.48 ± 0.96), width 5.84 ± 0.53 (4.46 ± 0.28). Palp 12.90 ± 0.60 (14.30 ± 0.77); leg I 38.86 ± 1.24 (44.61 ± 3.72); leg II 40.22 ± 1.03 (43.69 ± 3.83); leg III 37.32 ± 1.98 (38.56 ± 3.04); leg IV 43.87 ± 1.50 (45.82 ± 2.72). Diameters of AME 0.43 ± 0.06 (0.40 ± 0.02), ALE 0.31 ± 0.02 (0.29 ± 0.01), PME 0.58 ± 0.06 (0.54 ± 0.04), PLE 0.60 ± 0.03 (0.59 ± 0.04); Clypeus 1.17 ± 0.08 (0.98 ± 0.08). Chelicera length 3.42 ± 0.15 (3.21 ± 0.23). All the specimens have three promarginal and four retromarginal teeth on both chelicerae except the Holotype male which has asymmetrical number of the cheliceral marginal teeth. Endite length 2.61 ± 0.23 (2.49 ± 0.18), width 1.50 ± 0.07 (1.17 ± 0.04). Labium length 1.45 ± 0.09 (1.18 ± 0.01), width 1.68 ± 0.13 (1.43 ± 0.19). Sternum length 3.41 ± 0.09 (3.38 ± 0.36), width 3.79 ± 0.24 (3.53 ± 0.44). Male D. gregoric sp. nov. has two different coloration morphs: the dark morph (Fig. 22D) has only light colored but indistinct carapace edge that fades in the anterior carapace and in the lateral abdomen; and the white banded morph (Figs. 22I, 25A) with very distinct, but thin, white lateral bands that do not expand to the edge of the carapace. We did not find such color variations in the female D. gregoric sp. nov. Considering the white banded morph is in general less abundant in the surveyed regions and we found only two females, it is possible that the coloration variations also occur in female D. gregoric sp. nov.

Natural history. Large sized wandering spider but slightly smaller, albeit not significant (see Result) than the other two closely related Dolomedes species. Known to inhabit only the steep forest stream with several waterfalls along trail Chute Sacree, Parc national d’Andasibe-Mantadia. Active at night, found on the edge between roots or tree trunks that grow into the water (Figs. 22C, 22D and 22I). Population size much smaller than that of D. bedjanic sp. nov. with which it cohabits.

Etymology. The species is named after our colleague Matjaž Gregorič who organized the field trip and contributed to the discovery of this species. The species epithet is a noun in apposition.

Distribution. Known only from the type localities (see Fig. 16B).

Dolomedes bedjanic Yu & Kuntner sp. nov.

(Figs. 16C, 18C, 18H, 19C, 20C, 20F, 21C, 22E, 22F, 27, 28)

Figure 27 Male Dolomedes bedjanic sp. nov. (Holotype, USNMENT01580827).

(A) Dorsal view. (B) Anterior view. (C and D) Left palp: (C) Ventral view; (D) retrolateral view showing the length ratio between cymbium and tibia. (E and F) Left palp: (E) Ventral view; (F) retrolateral view showing anatomical details.

Figure 28 Female Dolomedes bedjanic sp. nov. (Paratype, USNMENT01580828).

(A) Dorsal view. (B) Anterior view. (C and D) Epigynum: (C) Ventral view; (D) dorsal view.

ZooBank id: urn:lsid:zoobank.org:act:081004EB-ED43-4BA1-9AB9-C7DCE95BB338

Holotype male. MADAGASCAR: Antsiranana Province: the 4th stream below Mantella Camp, on the trail toward Mandena village, Parc National de Marojejy (14°26′38.4″S, 49°47′0.9″E, 366 m), 25 III 2022, KPY leg., USNMENT01580827 (USNM).

Paratype female. Same collecting locality as the Holotype, 29 III 2022, KPY leg., USNMENT01580828 (USNM).

Other material examined. MADAGASCAR: Antsiranana Province: one female and one male, the 1st stream below Mantella Camp, on the trail toward Mandena village, Parc National de Marojejy (14°26′22.4″S, 49°46′38.8″E, 463 m), 25–30 III 2022, KPY leg., KPARA00129, 00166 (NIB); two females and one juvenile, the 3rd stream below Mantella Camp, on the trail toward Mandena village, Parc National de Marojejy (14°26′29.5″S, 49°46′48.2″E, 412 m), 25–30 III 2022, KPY leg., KPARA00131, 00154 (NIB); one female, same collecting information as the Holotype, KPARA00132 (NIB); one female, the 5th stream below Mantella Camp, on the trail toward Mandena village, Parc National de Marojejy (14°26′47.3″S, 49°47′6.5″E, 342 m), 25 III 2022, KPY leg., KPARA00133 (NIB); two females and five juveniles, the stream on the trail toward Cascade de Humbert, Parc National de Marojejy (14°26′3.72″S, 49°46′21.8″E, 546 m), 28 III 2022, KPY leg., KPARA00144–146, 00159–162 (NIB). Toamasina Province: three females and two males, the streams around Lac Vert, Reserve Analamazoatra (18°56′14.2″S, 48°25′12.5″E, 939 m), 4–8 IV 2022, KPY leg., KPARA00192, 00194, 00227, 00232–233 (NIB); one juvenile, the stream around Orchid Lake, Parc Mitsinjo (18°55′58.2″S, 48°24′48.9″E, 935 m), 7 IV 2022, KPY leg., KPARA00226 (NIB); two males, the slow flowing stream and swamps along trail Kalanoro, next to Vakona Lodge (18°53′16.9″S, 48°26′4.5″E, 987 m), 10 IV 2022, KPY leg., KPARA00234–235 (NIB); four females and one male, the stream along the trail Chute Sacree, Parc national d’Andasibe-Mantadia (18°49′30.7″S, 48°26′5.3″E, 976 m), 6 & 12 IV 2022, KPY leg. KPARA00202, 00247, 00251–252, 00255 (NIB).

Diagnosis. Male D. bedjanic sp. nov. differs from the other two Dolomedes species of the “Kalanoro” group by i) the long palp that is twice longer than the carapace width (Fig. 2H); ii) the RTA positioned near dorsally (Figs. 18H, 27D and 27F) with only a sharp ventral lobe (Fig. 21C); iii) the Cy shorter than the palp tibia (Fig. 5B); iii) the MA expands gradually from a relatively broader base to the middle (Figs. 18C, 27C and 27E); and v) the short Eb with a wide base (Figs. 18C, 19C, 27C and 27E). Female D. bedjanic sp. nov. can be separated from the other two Dolomedes species of the “Kalanoro” group by i) the wide MF without a horn extension (Figs. 20F, 28C); and ii) the short CD with less than a loop (Figs. 20C, 28D).

Description. Male (Holotype, USNMENT01580827). Total length 15.89: carapace length 8.69, width 7.88, anterior height 3.03, posterior height 4.12; abdomen length 7.20, width 4.02. Length of palp and legs: palp 17.07 (7.09, 2.65, 3.72, 3.61); leg I 47.61 (12.66, 4.63, 12.65, 11.80, 5.87); leg II 46.92 (12.55, 4.69, 12.20, 11.44, 6.04); leg III 41.88 (11.44, 4.19, 10.62, 10.54, 5.09); leg IV 43.35 (12.28, 4.38, 12.22, 12.94, NA). Leg formula 1243. Carapace pear-shaped, brownish with dense black short setae that form series of black radial markings pointing towards the distinct fovea (Fig. 27A). A pair of triangular black markings positioned anterior to the fovea (Fig. 27A). Edge of the carapace with sparse greyish setae that form indistinct light-colored bands (Fig. 27A). Eight eyes ringed with black (Fig. 27B). Eyes arranged in two rows. AER weakly recurved while PER strongly recurved. MOA dark brown with several black setae (Fig. 27B). Diameters of AME 0.39, ALE 0.29, PME 0.51, PLE 0.59; MOA length 1.09, anterior width of MOA 0.89, posterior width of MOA 1.34; interval of AMEs 0.14, interval of PMEs 0.23, interval between AME and ALE 0.14, interval between PME and PLE 0.42. Clypeus 1.14, brown covered with dark short setae (Fig. 27B). Chelicera length 3.5, chestnut-brown covered with black long setae (Fig. 25B). The right chelicera with four promarginal and six retromarginal teeth whereas the left chelicera with three and four teeth respectively. Both fangs and marginal teeth black. Endite length 2.68, width 1.27; labium length 1.51, width 1.54; sternum near round, length 3.63, width 3.59; all endites, labium, and sternum yellowish-brown. Abdomen long oval with a distinct yellowish cardiac mark; dorsum with dense, dark brown short setae (Fig. 27A). Yellowish setae cover the lateral and the dorsal abdomen that forms indistinct light-colored lateral bands and dorsal patches (Fig. 27A). Venter abdomen brown. Legs brown, covered with dark blackish setae in different densities, regions with sparser setae form light-colored linear markings. Palp long, twice longer than the carapace width (Fig. 2H). Tibia with a highly sclerotized RTA positioned near dorsally (Figs. 18H, 27D and 27F) with only a sharp ventral lobe (Fig. 21C). Cy shorter that the palp tibia (Fig. 5B). Basal retrolateral edge of the ventral Cy with an oval shaped BCA (Figs. 18C, Figs. 27C and 27E). T sclerotized with a membranous upper edge, the prolateral side attached to the DTP and the retrolateral side attached to the membranous Co. T + DTP + Co forms a “U” shaped tegular ring (Figs. 18C, 27C and 27E). A highly sclerotized Sa sits at the lower center of the ring, attached to the T (Figs. 18C, 27C and 27E). MA sits retrolaterally to the Sa, hook-shaped and expands gradually from the relatively broad base to the middle (Figs. 18C, 27C and 27E). Fu hook shaped, retrolateral side with a small but distinct lateral lobe (Figs. 18C, 27C and 27E). Ventral edge of the Fu folded and forms a groove that contains the short, broad-based, and curved Eb (Figs. 18H, 19C, 27D and 27F). LA trapezoid with a blunt and broad ventral tip (Fig. 19C). All Fu, Eb, and LA originated from the DST (Fig. 19C).

Female (Paratype, USNMENT01580828). Total length 21.55: carapace length 10.66, width 9.44, anterior height 3.62, posterior height 4.03; abdomen length 10.89, width 7.58. Length of palp and legs: palp 13.58 (4.67, 1.96, 2.81, 4.14); leg I 40.21 (10.67, 4.27, 10.85, 9.34, 5.08); leg II 41.35 (11.46, 4.53, 10.88, 9.38, 5.10); leg III 38.72 (10.90, 4.33, 9.53, 9.40, 4.56); leg IV 45.58 (11.70, 4.29, 11.38, 12.17, 6.04). Leg formula 4213. Diameters of AME 0.50, ALE 0.33, PME 0.63, PLE 0.65; MOA length 1.38, anterior width of MOA 1.11, posterior width of MOA 1.61; interval of AMEs 0.17, interval of PMEs 0.27, interval between AME and ALE 0.18, interval between PME and PLE 0.61. Clypeus 1.37. Chelicera length 4.47. Both chelicerae with three promarginal and four retromarginal teeth. Endite length 3.15, width 1.67; labium length 1.74, width 1.95; sternum near round, length 4.09, width 4.45. Female similar to the male, but larger, darker in the coloration, and with relatively shorter legs. A series of white spots and lines distributed on the dorsal abdomen (Figs. 28A, 28B). Epigynum pentagon shaped, longer than wide and highly sclerotized; divided into two LLs by the wide rectangular MF with two small and distinct membranous windows (Figs. 20F, 28C). CD short, less than one loop and connected to the BS anteriorly (Figs. 20C, 28D). AB small, horn-shaped and anteriorly positioned (Figs. 20C, 28D). FD begins with a coiled and tubular part; ended with a triangular flake (Figs. 20C, 28D).

Variation. Given as variation of 14 females followed by variation of seven males in parentheses. Total length 21.21 ± 2.10 (16.99 ± 1.11): carapace length 10.61 ± 0.84 (8.59 ± 0.44), width 9.34 ± 0.86 (7.66 ± 0.38), anterior height 3.52 ± 0.32 (3.03 ± 0.16), posterior height 4.04 ± 0.45 (3.78 ± 0.26); abdomen length 10.60 ± 1.39 (8.40 ± 0.88), width 6.57 ± 1.38 (4.75 ± 0.68). Palp 15.63 ± 1.29 (16.33 ± 0.68); leg I 45.28 ± 3.75 (46.03 ± 2.19); leg II 45.86 ± 3.63 (44.60 ± 2.14); leg III 42.41 ± 3.53 (39.15 ± 2.15); leg IV 49.16 ± 4.05 (44.32 ± 1.08). Diameters of AME 0.45 ± 0.04 (0.39 ± 0.02), ALE 0.31 ± 0.02 (0.27 ± 0.02), PME 0.60 ± 0.05 (0.52 ± 0.02), PLE 0.64 ± 0.06 (0.57 ± 0.03); Clypeus 1.47 ± 0.16 (1.12 ± 0.05). Chelicera length 4.39 ± 0.43 (3.45 ± 0.21). All the specimens have three promarginal and four retromarginal teeth on both chelicerae except the Holotype male which has asymmetrical number of the cheliceral marginal teeth. Endite length 3.16 ± 0.29 (2.54 ± 0.16), width 1.67 ± 0.09 (1.29 ± 0.03). Labium length 1.76 ± 0.18 (1.42 ± 0.08), width 1.90 ± 0.18 (1.46 ± 0.09). Sternum length 4.18 ± 0.42 (3.55 ± 0.17), width 4.43 ± 0.43 (3.63 ± 0.22). Although D. bedjanic sp. nov. has relatively larger population in the investigated regions compared to the other two species of the “Kalanoro” group, no white banded morph has been found.

Natural history. Large sized wandering spider. Inhabits water bodies, both flowing or standing, with dense canopy coverage. Active at night, found on surfaces near or on water (Figs. 22C, 22D). One individual (KPARA00251) was sitting on a riverside tree trunk away from the water. An individual (KPARA00202) was found resting in holes on the river bank during day time.

Etymology. The species is named after our colleague Matjaž Bedjanič who participated in the aquatic collecting work and contributed to the discovery of this species. The species epithet is a noun in apposition.

Distribution. Northern and eastern humid forests of Madagascar (see Fig. 16C).

Species group “Hydatostella”

Diagnosis. Dolomedes species of the “Hydatostella” species group can be distinguished from all the other known Dolomedes, except the Madagascar Dolomedes species of the group “Kalanoro”, by the combination of the following characters:

Male: The presence of a lateral lobe at the retrolateral edge of the Fu in the ventral view (Fig. 18, black arrows; see species descriptions for the structure in D. rotundus).

Female: i) epigynum longer than wide, without lateral extensions at the posterior edge (Figs. 20F–20J); ii) two fully separated, medially positioned, relatively small, and triangular or square-shaped MF windows (Figs. 20F–20J); iii) CD wider than the first loop of the FD, or in similar width (Figs. 20A–20E); and iv) FD simply and vertically coiled without any contrary flexure or horizontal spiral (Figs. 20A–20E).

Dolomedes of the group “Hydatostella” can be distinguished from those of the “Kalanoro” group by i) the smaller body size, ii) the relatively shorter appendages, and iii) the dark brownish to blackish coloration with very distinct white lateral bands that expand to the edge of the carapace.

Composition. Dolomedes hydatostella sp. nov., D. rotundus sp. nov.

Dolomedes hydatostella Yu & Kuntner sp. nov.

(Figs. 16D, 18D, 18I, 19D, 20D, 20I, 21D, 22J, 22K, 29, 30)

Figure 29 Male Dolomedes hydatostella sp. nov. (Holotype, USNMENT01580829).

(A) Dorsal view. (B) Anterior view. (C and D) Left palp: (C) Ventral view; (D) retrolateral view showing the length ratio between cymbium and tibia. (E and F) Left palp: (E) Ventral view, the black arrow showing the lateral lobe on the fulcrum; (F) retrolateral view showing anatomical details.

Figure 30 Female Dolomedes hydatostella sp. nov. (Paratype, USNMENT01580830).

(A) Dorsal view. (B) Anterior view. (C and D) Epigynum: (C) Ventral view; (D) dorsal view.

ZooBank id: urn:lsid:zoobank.org:act:6C524835-816A-42F8-8700-BF1A597F8ED9

Holotype male. MADAGASCAR: Antsiranana Province: the 2th muddy stream and swamp below Mantella Camp, on the trail toward Mandena village, Parc National de Marojejy (14°26′26.8″S, 49°46′44.6″E, 433 m), 30 III 2022, KPY leg., USNMENT01580829 (USNM; collected as sub adult).

Paratype female. Same collecting information as the Holotype, USNMENT01580830 (USNM).

Other material examined. Four females and one male, same collecting information as the Holotype, KPARA00157–158, 00163–164, 00258 (NIB).

Diagnosis. Male D. hydatostella sp. nov. differs from D. rotundus sp. nov. by i) the similar sized RTA dorsal and ventral lobes (Fig. 21D); ii) the straight Eb retrolateral arc, in the ventral view (Figs. 18D, 29C and 29E); and iii) the LA with a distinct narrow base (Fig. 19D). Female D. hydatostella sp. nov. differs from D. rotundus sp. nov. by i) the pentagon shaped epigynum (Figs. 20I, 30C); ii) the relatively longer posterior MF (Figs. 20I, 30C); and iii) the shorter EF (Figs. 20I, 30C).

Description. Male (Holotype, USNMENT01580829). Total length 11.63: carapace length 6.23, width 5.88, anterior height 2.19, posterior height 2.49; abdomen length 5.40, width 2.91. Length of palp and legs: palp 9.49 (3.69, 1.53, 1.85, 2.42); leg I 26.82 (7.11, 2.96, 7.10, 6.27, 3.38); leg II 27.21 (7.42, 2.92, 7.10, 6.40, 3.37); leg III 24.53 (6.91, 3.83, 6.23, 5.71, 2.85); leg IV 29.29 (7.92, 3.07, 7.21, 7.37, 3.72). Leg formula 4213. Carapace pear-shaped, light brown with dense black short setae (Fig. 29A). Fovea distinct, and extends posteriorly from the center (Fig. 29A). A pair of triangular black spots positioned anteriorly to the fovea. Lateral carapace with dense white setae that form distinct wide lateral bands (Fig. 29A). Lateral bands wider in the first half of the carapace and gradually shrink to the posterior carapace (Fig. 29A). Eight eyes ringed with black. Eyes arranged in two rows. AER weakly recurved while PER strongly recurved. MOA brown with several black setae (Fig. 29B). Diameters of AME 0.35, ALE 0.20, PME 0.42, PLE 0.44; MOA length 0.78, anterior width of MOA 0.71, posterior width of MOA 1.04; interval of AMEs 0.10, interval of PMEs 0.18, interval between AME and ALE 0.12, interval between PME and PLE 0.3. Clypeus 0.74, light yellowish covered with dark short setae. Chelicera length 2.28, light brown covered with black long setae (Fig. 29B). Both chelicerae with three promarginal and four retromarginal teeth. Both fangs and marginal teeth chestnut brown. Endite length 1.84, width 1.00; labium length 0.93, width 1.00; sternum near round, length 2.36, width 2.65; all endites, labium, and sternum yellowish-brown. Abdomen long oval with a distinct and greyish-brown cardiac mark; dorsum with dense, dark brown short setae with a series of white spots (Fig. 29A). Lateral abdomen covered with white short setae that form distinct and irregular lateral bands and patches (Fig. 29A). Venter abdomen brownish. Legs light brown, covered with dark blackish setae in different densities, regions with sparser setae form light-colored linear markings (Fig. 29A). Palp tibia with a highly sclerotized RTA (Figs. 18I, 29D and 29F) divided into two sharp lobes that are in similar size (Fig. 21D). Basal retrolateral edge of the ventral Cy with an oval BCA (Fig. 18D, 29C and 29E). T sclerotized with a membranous upper edge, the prolateral side attached to the DTP and the retrolateral side attached to the membranous Co (Figs. 18D, 29C and 29E). T + DTP + Co forms a “U” shaped tegular ring (Figs. 18D, 29C and 29E). A highly sclerotized Sa sits at the lower center of the ring, attached to the T (Figs. 18D, 29C and 29E). MA sits retrolaterally to the Sa, hook-shaped and expands gradually from the narrow base (Figs. 18D, 29C and 29E). Fu hook shaped, retrolateral side with a small but distinct lateral lobe (Figs. 18D, 29C and 29E). Ventral edge of the Fu folded and forms a groove that contains the short, broad-based, and curved Eb (Figs. 18I, 19D, 29D and 29F). From the ventral view, retrolateral arc of the Eb straight (Figs. 18D, 29C and 29E). LA trapezoid with round tips and a distinct narrow base (Fig. 19D). All Fu, Eb, and LA originated from the DST (Fig. 19D).

Female (Paratype, USNMENT01580830). Total length 17.09: carapace length 7.18, width 6.58, anterior height 2.57, posterior height 3.15; abdomen length 9.91, width 6.23. Length of palp and legs: palp 9.11(2.96, 1.37, 1.96, 2.82); leg I 23.69 (6.80, 2.91, 6.56, 5.11, 2.31); leg II 24.70 (7.36, 2.96, 6.70, 5.24, 2.44); leg III 23.07 (6.75, 2.74, 6.01, 5.10, 2.47); leg IV 27.65 (7.78, 2.83, 7.08, 6.92, 3.04). Leg formula 4213. Diameters of AME 0.38, ALE 0.24, PME 0.44, PLE 0.50; MOA length 0.94, anterior width of MOA 0.79, posterior width of MOA 1.18; interval of AMEs 0.11, interval of PMEs 0.24, interval between AME and ALE 0.15, interval between PME and PLE 0.35. Clypeus 0.83. Chelicera length 2.82. The right chelicera with three promarginal and five retromarginal teeth whereas the left chelicera with three and four teeth respectively. Endite length 1.87, width 1.10; labium length 0.98, width 1.31; sternum near round, length 2.79, width 3.02. Female similar to male, but larger, much darker in the coloration, and with relatively shorter legs that have additional dense white setae on the metatarsi (Figs. 30A, 30B). Epigynum pentagon shaped, highly sclerotized, and longer than wide; divided into two LLs by the near rectangular MF with two small and distinct membranous windows and a relatively larger posterior part (Figs. 20C, 30C). EF short and does not extend to near the anterior margin of the epigynum (Figs. 20I, 30C). CD short with less than one loop (Figs. 20D, 30D). AB small, horn-shaped and laterally positioned (Figs. 20D, 30D). FD begins with a coiled and tubular part and ends with a thin flake (Figs. 20D, 30D).

Variation. Given as variation of five females followed by variation of two males in parentheses. Total length 15.54 ± 1.41 (12.58 ± 1.34): carapace length 6.95 ± 0.34 (6.38 ± 0.21), width 6.43 ± 0.28 (6.03 ± 0.21), anterior height 2.61 ± 0.25 (2.26 ± 0.09), posterior height 2.91 ± 0.28 (2.70 ± 0.30); abdomen length 8.59 ± 1.34 (6.21 ± 1.14), width 5.48 ± 0.83 (3.64 ± 1.03). Palp 9.17 ± 0.37 (9.74 ± 0.35); leg I 23.69 ± 1.16 (27.54 ± 1.02); leg II 24.50 ± 1.38 (27.88 ± 0.95); leg III 23.02 ± 0.96 (24.84 ± 0.43); leg IV 27.96 ± 1.22 (29.60 ± 0.43). Diameters of AME 0.36 ± 0.03 (0.28 ± 0.11), ALE 0.36 ± 0.02 (0.21 ± 0.01), PME 0.23 ± 0.03 (0.41 ± 0.01), PLE 0.45 ± 0.03 (0.44 ± 0.01); Clypeus 0.48 ± 0.10 (0.77 ± 0.02). Chelicera length 2.71 ± 0.13 (2.37 ± 0.12). Most of the specimens have three promarginal and four retromarginal teeth on both chelicerae while two specimens have asymmetrical number of the cheliceral marginal teeth. Endite length 1.86 ± 0.08 (1.81 ± 0.70), width 1.11 ± 0.05 (0.97 ± 0.34). Labium length 0.95 ± 0.09 (0.94 ± 0.42), width 1.22 ± 0.10 (1.01 ± 0.41). Sternum length 2.78 ± 0.17 (2.50 ± 0.84), width 3.01 ± 0.19 (2.76 ± 0.82).

Natural history. Median-sized wandering spider inhabiting shallow standing water bodies or very slow-flowing parts of streams with dense canopy coverage. Active at night, found holding on to leaf litter or grass emerging from the water surface (Fig. 22I).

Etymology. The species epithet “hydatostella” is a compound noun in apposition. The first half “hydatos-” is a Latinized Greek, “ὕδωρ (hydōr)”, refering to “water”. The second half “-stella” is a Latin word refering to “stars”. Together the species epithet means “stars on water”.

Distribution. Known only from the type locality (see Fig. 16D).

Dolomedes rotundus Yu & Kuntner sp. nov.

(Figs. 16D, 18E, 18J, 19E, 19F, 20E, 20J, 21E, 22L, 22M, 31, 32)

Figure 31 Male Dolomedes rotundus sp. nov. (Holotype, USNMENT01580831).

(A) Dorsal view. (B) Anterior view. (C and D) Left palp: (C) Ventral view; (D) retrolateral view showing the length ratio between cymbium and tibia. (E and F) Left palp: (E) Ventral view; (F) retrolateral view showing anatomical details.

Figure 32 Female Dolomedes rotundus sp. nov. (Paratype, USNMENT01580832).

(A) Dorsal view. (B) Anterior view. (C and D) Epigynum: (C) Ventral view; (D) dorsal view.

ZooBank id: urn:lsid:zoobank.org:act:03ADE51C-2B73-4F7F-B2BD-61755461912C

Holotype male. MADAGASCAR: Toamasina Province: the slow flowing stream and swamps along trail Kalanoro, next to Vakona Lodge (18°53′16.9″S, 48°26′4.5″E, 987 m), 10 IV 2022, KPY leg., USNMENT01580831 (USNM).

Paratype female. Same collecting information as the Holotype, USNMENT01580832 (USNM).

Other material examined. MADAGASCAR: Toamasina Province: two juveniles, slow flowing part of the streams around Lac Vert, Reserve Analamazoatra, Toamasina Province (18°56′14.2″S, 48°25′12.5″E, 939 m), 4–8 IV 2022, KPY leg., KPARA00229–230 (NIB); two females, four males, and three juveniles, same collecting information as the Holotype, KPARA00236–239, 00241, 00243, 00244 (NIB); one male, forêt Analalava, Foulpointe, Tamatave, I 1995, Pauly A. leg., MT_207084 (RMCA).

Diagnosis. Male D. rotundus sp. nov. differs from D. hydatostella sp. nov. by i) the round dorsal RTA lobe which is larger than the ventral lobe (Fig. 21E); ii) the bent retrolateral arc of the Eb, in the ventral view (Figs. 18E, 31C and 31E); and iii) the LA with a broad base (Figs. 19E and 19F). Female D. rotundus sp. nov. differs from D. hydatostella sp. nov. by i) the triangular or round-shaped epigynum (Figs. 20J, 32C); ii) the relatively shorter posterior MF (Figs. 20J, 32C); and iii) the longer EF extending to near the anterior edge of the epigynum (Figs. 20J, 32C).

Description. Male (Holotype, USNMENT01580831). Total length 13.15: carapace length 6.47, width 5.80, anterior height 2.19, posterior height 2.85; abdomen length 6.68, width 4.38. Length of palp and legs: palp 9.88 (3.93, 1.55, 1.98, 2.42); leg I 26.74 (7.25, 2.90, 7.10, 6.07, 3.42); leg II 26.80 (7.49, 2.85, 7.12, 6.09, 3.25); leg III 24.56 (7.08, 2.67, 6.20, 5.72, 2.89); leg IV 29.12 (7.94, 2.83, 7.08, 7.41, 3.86). Leg formula 4213. Carapace pear-shaped, light yellowish with dense black short setae (Fig. 31A). Fovea distinct, and extends posteriorly from the center (Fig. 31A). A pair of triangular black spots positioned anteriorly to the fovea (Fig. 31A). First half of the lateral carapace with dense white setae that form distinct wide lateral patches (Fig. 31A). The dense white setae coverages distinctly sparser at the middle of the carapace edge, and become thin and fading (Fig. 31A). Eight eyes ringed with black. Eyes arranged in two rows. AER weakly recurved while PER strongly recurved (Fig. 31B). MOA brownish with several black setae (Fig. 31B). Diameters of AME 0.37, ALE 0.24, PME 0.41, PLE 0.44; MOA length 0.90, anterior width of MOA 0.74, posterior width of MOA 1.01; interval of AMEs 0.11, interval of PMEs 0.13, interval between AME and ALE 0.10, interval between PME and PLE 0.38. Clypeus 0.67, light yellowish covered with sparse dark short setae (Fig. 31B). Chelicera length 2.56, yellowish covered with black long setae (Fig. 31B). Both chelicerae with three promarginal and four retromarginal teeth. Fangs and marginal teeth dark brown. Endites length 1.73, width 0.97; labium length 0.89, width 1.00; sternum near round, length 2.50, width 2.88; Endites, labium, and sternum light yellowish (Fig. 31A). Abdomen long oval with a distinct and greyish-brown cardiac mark; dorsum with dense, dark brown short setae with a series of white spot (Fig. 31A). Lateral abdomen covered with white short setae that form distinct and irregular lateral bands and patches (Fig. 31A). Venter abdomen brownish. Legs brownish covered with blackish short setae in different densities, regions with sparser setae form light-colored linear markings (Fig. 31A). Palp tibia with a highly sclerotized RTA (Figs. 18J, 21E, 31D and 31F) divided into a sharp ventral lobe and a larger, round-tipped dorsal lobe (Fig. 21D). Basal retrolateral edge of the ventral Cy with an oval shaped BCA (Figs. 18E, 31C and 31E), T sclerotized with a membranous upper edge, the prolateral side attached to the DTP and the retrolateral side attached to the membranous Co (Figs. 18E, 31C and 31E). T + DTP + Co forms a “U” shaped tegular ring (Figs. 18E, 31C and 31E). A highly sclerotized Sa sits at the lower center of the tegular ring, attached to the T (Figs. 18E, 31C and 31E). MA sits retrolaterally to the Sa, hook-shaped and expands gradually from the narrow base (Figs. 18E, 31C and 31E). Fu hook shaped, retrolateral side with a small lateral lobe (Figs. 18E, 31C and 31E). Ventral edge of the Fu folded and forms a groove that contains curved Eb (Figs. 18J, 19E–19F, 31D and 31F). In the ventral view, retrolateral arc of the Eb bent (Figs. 18E, 31C and 31E). LA trapezoid with round tips with a broad base (Figs. 19E and 19F). All Fu, Eb, and LA originated from the DST (Figs. 19E and 19F).

Female (Paratype, USNMENT01580832). Total length 16.05: carapace length 6.56, width 6.00, anterior height 2.21, posterior height 2.47; abdomen length 9.49, width 6.36. Length of palp and legs: palp 8.36 (2.81, 1.26, 1.82, 2.47); leg I 21.44 (6.21, 2.58, 5.93, 4.47, 2.25); leg II 22.26 (6.74, 2.59, 5.97, 4.65, 2.31); leg III 20.44 (6.10, 2.53, 5.42, 4.46, 1.93); leg IV 25.14 (7.08, 2.82, 6.43, 6.24, 2.57). Leg formula 4213. Diameters of AME 0.29, ALE 0.23, PME 0.43, PLE 0.43; MOA length 0.87, anterior width of MOA 0.70, posterior width of MOA 1.06; interval of AMEs 0.12, interval of PMEs 0.21, interval between AME and ALE 0.13, interval between PME and PLE 0.41. Clypeus 0.74. Chelicera length 2.71. Both chelicerae with three promarginal and four retromarginal teeth. Endite length 1.69, width 1.02; labium length 0.89, width 1.11; sternum near round, length 2.61, width 2.83. Female similar to male, but larger, darker the in coloration, and with relatively shorter legs that have additional dense gray setae on the metatarsi (Figs. 32A, 32B). Epigynum round or triangle shaped, highly sclerotized and longer than wide; divided into two LLs by the near oval MF with two small and distinct membranous windows and a relatively small posterior part (Figs. 20J, 32C). EF long, extends to near the anterior edge of the epigynum (Figs. 20J, 32C). CD short with less than one loop (Figs. 20E, 32D). AB small, knob-shaped, and laterally positioned (Figs. 20E, 32D). FD begins with a coiled and tubular part and ends with a thin flake (Figs. 20E, 32D).

Variation. Given as variation of three females followed by variation of six males in parentheses. Total length 14.68 ± 1.23 (12.11 ± 1.10): carapace length 6.77 ± 0.27 (6.05 ± 0.45), width 6.20 ± 0.23 (5.56 ± 0.40), anterior height 2.28 ± 0.16 (2.13 ± 0.12), posterior height 2.64 ± 0.16 (2.74 ± 0.15); abdomen length 7.91 ± 1.38 (6.07 ± 0.67), width 5.08 ± 1.16 (3.82 ± 0.72). Palp 8.96 ± 0.35 (12.56 ± 0.54); leg I 22.30 ± 1.09 (25.49 ± 1.54); leg II 23.19 ± 0.97 (25.72 ± 1.70); leg III 21.61 ± 1.19 (23.30 ± 1.45); leg IV 26.45 ± 1.62 (27.41 ± 1.56). Diameters of AME 0.33 ± 0.03 (0.31 ± 0.04), ALE 0.22 ± 0.01 (0.22 ± 0.02), PME 0.43 ± 0.01 (0.40 ± 0.02), PLE 0.45 ± 0.02 (0.44 ± 0.02); Clypeus 0.48 ± 0.10 (0.77 ± 0.02). Chelicera length 2.61 ± 0.15 (2.42 ± 0.14). Most of the specimens have three promarginal and four retromarginal teeth on both chelicerae while one specimen has asymmetrical number of the cheliceral marginal teeth. Endite length 1.82 ± 0.13 (1.66 ± 0.18), width 1.04 ± 0.03 (0.89 ± 0.10). Labium length 0.96 ± 0.07 (0.84 ± 0.10), width 0.96 ± 0.07 (0.84 ± 0.10). Sternum length 2.64 ± 0.03 (2.46 ± 0.21), width 2.89 ± 0.08 (2.74 ± 0.24). The specimen from Tamatave (MT_207084) has a slightly different LA with a sharp dorsal tip (Fig. 19F).

Natural history. Median-sized wandering spider inhabiting habitats similar to D. hydatostella sp. nov. Active at night, found floating on water, or holding on to leaf litter or vegetation emerging from the water surface (Figs. 22L, 22M).

Remark. Specimens of the “Hydatostella” group from the north and the east of the island are very similar and are nearly identical in the habitus. However, the differences between the two populations are constant and receive strong support in both morphological (see Results) and phylogenetic analyses (Fig. 15, clades A and B). The poor dispersal abilities and geological separation can as well explain the limitation of gene flow between the populations (see Results and Discussion). We hence separate D. rotundus sp. nov. as a new species.

Etymology. The species epithet is a masculine adjective, referring to the major diagnostic characteristics of the species: round shaped female epigynum and male RTA dorsal lobe.

Distribution. Known from limited areas of eastern humid forests of Madagascar (see Fig. 16D).

Supplemental Information

Supplemental Information 1 Retrolateral view of the right distal sclerotized tube of male Dolomedes kalanoro da Silva & Griswold, 2013 (KPARA00185), showing the examples of landmark positions.

(A) Embolus, the gray lines represent the reference lines for landmarks plotting, the red line represent the diameter of embolic ring (De). (B) Fulcrum. (C) Lateral subterminal apophysis. The black arrow showing the dorsal direction.

Supplemental Information 2 The left palp of male Dolomedes kalanoro da Silva & Griswold, 2013 (KPARA00185), showing examples of landmark positions.

(A) Median apophysis, ventral view. (B) Embolus, ventral view. (C) Fulcrum, dorsal view. (D) retrolatateral tibial apophysis, posterolateral view.

Supplemental Information 3 The epigynum of Dolomedes kalanoro da Silva & Griswold, 2013 (KPARA00193), showing examples of landmark positions.

A) Epigynal margin, ventral view. (B) Epigynal middle field, ventral view. (C) Vulva arrangement, dorsal view.

Supplemental Information 4 Examples of the four habitat categories used in the study.

(A) The pond at the countryside of Sambava, represents standing water with an open canopy. (B) Understory swamps at Parc National de Marojejy, represent standing water with dense canopy. (C) The river along Circuit Tsakoka, Parc National d’Andasibe-Mantadia, represents flowing water with an open canopy. (D) The stream at Parc National de Marojejy, represents flowing water with a dense canopy.

Supplemental Information 5 The Mantispidae larva found in the epigastric furrow of a female Dolomedes bedjanic sp. nov. (KPARA00144).

(A–B) Parasitized epigynum: (A) Ventral view; (B) dorsal view, showing the deformed vulva. (C–D) The Mantispidae larva: (C) Dorsal view; (D) lateral view.

Supplemental Information 6 Descriptions of the landmarks selected within each structure.

RTA: retrolateral tibial apophysis, LA: lateral subterminal apophysis.

Supplemental Information 7 GenBank accession numbers of the COI sequences included in this study.

Supplemental Information 8 Results of the one-way analyses of variance showing the measurements of the six selected structures are significantly different among the five morphospecies in both sexes.

De: diameter of embolic ring; Df: degrees of freedom; SS: sum of square; MS: mean square; significance threshold: p-value < 0.05

Supplemental Information 9 Pairwise comparisons in measurements of the six selected structures between all pair combinations of the five morphospecies.

De: diameter of embolic ring; diff: difference between means of the two groups; lwr: lower end point of 95% confident interval; upr: upper end point of 95% confident interval; significance threshold: p-value < 0.05; significant p-value were in bold.

Supplemental Information 10 Proportion and cumulative variances in the shape components of each selected structure explained by the first five principal component axes (PC).

RTA: retrolateral tibial apophysis, LA: lateral subterminal apophysis.

Supplemental Information 11 List of collecting information and the deposition of the specimens examined in this study.

NIB: National Institute of Biology, Ljubljana, Slovenia; RMCA: Royal Museum for Central Africa, Tervuren, Belgium; SMF: Senckenberg Natural History Museum, Frankfurt, Germany; USNM: National Museum of Natural History, Smithsonian Institution, Washington, DC, USA; Sub: Sub-adult; F: Female; M: Male; J: Juvenile

Supplemental Information 12 Raw data and code for analyses performed in this study.

The raw measurements, landmark coordinates, and the R code for morphological analyses. An aligned COI matrix is also included.

We sincerely thank Tiana Vololona, Rhina Harin’ Hala Rasolondalao, and the staff at Madagascar’s Institute for the Conservation of Tropical Ecosystems (MICET) for their help in the transportation and obtaining the research permits. We would like to express our thanks to the local students Annie Rasoanoeliarimanana, and Jeremia Ravelojaona for their help in the field. We are thankful to the local guides and the porters who supported our fieldwork in the remote forests. We acknowledge our colleagues Matjaž Gregorič and Matjaž Bedjanič for their help in organizing and executing the field trip. We thank Peter Jäger at SMF and Arnaud Henrard at RMCA for loaning the specimens, Peter Michalik for the suggestions in the integrative taxonomy, and Rok Kuntner for etymological advice. Finally, this article has much improved through a constructive peer review for which we thank Petra Sierwald, Danniella Sherwood, Marc Domènech and an anonymous colleague.

Abbreviations

AB accessory bulb

AER anterior eye row

AME anterior median eye

ALE anterior lateral eye

BCA basal cymbium apophysis

BS base of spermathecae

Co conductor

CD copulatory duct

Cy cymbium

COp copulatory opening

DTP distal tegular projection

DST distal sclerotized tube of apical division

DTA distal tegular apophysis

Eb embolus

EF epigynal fold

Fu fulcrum

FD fertilization duct

HS head of spermathecae

LA lateral subterminal apophysis

LL lateral lobe

MA median apophysis

MF middle field

MOA median ocular region

PER posterior eye row

PME posterior median eye

PLE posterior lateral eye

RTA retrolateral tibial apophysis

Sa saddle

T tegulum

Additional Information and Declarations

Competing Interests

Author Contributions

Field Study Permissions

DNA Deposition

Data Availability

New Species Registration

The authors declare that they have no competing interests.

Kuang-Ping Yu conceived and designed the experiments, performed the experiments, analyzed the data, prepared figures and/or tables, authored or reviewed drafts of the article, and approved the final draft.

Matjaž Kuntner conceived and designed the experiments, performed the experiments, authored or reviewed drafts of the article, and approved the final draft.

The following information was supplied relating to field study approvals (i.e., approving body and any reference numbers):

Direction of Protected Airs, Renewable Nature Resources and Ecosystems

The following information was supplied regarding the deposition of DNA sequences:

The COI sequences used in the study are available in the Supplemental File and at GenBank: OR284697 to OR284754.

The following information was supplied regarding data availability:

The details in the methodology, raw data matrices and R codes for morphometric analyses, the list of specimens and their deposition, as well as the aligned COI matrix, are available in the Supplemental Files, tables, and figures.

The following information was supplied regarding the registration of a newly described species:

Publication LSID: urn:lsid:zoobank.org:pub:C9091268-EC61-41CD-A20C-5C7DC08DAD46.

Dolomedes bedjanic: urn:lsid:zoobank.org:act:081004EB-ED43-4BA1-9AB9-C7DCE95BB338.

Dolomedes gregoric: urn:lsid:zoobank.org:act:075EEA19-43F1-4B65-97CC-5E014CE4D22C.

Dolomedes hydatostella: urn:lsid:zoobank.org:act:6C524835-816A-42F8-8700-BF1A597F8ED9.

Dolomedes rotundus: urn:lsid:zoobank.org:act:03ADE51C-2B73-4F7F-B2BD-61755461912C.

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
