# Peer review of "Discovering unknown Madagascar biodiversity: integrative taxonomy of raft spiders (Pisauridae: Dolomedes)"

_PeerJ, doi:10.7717/peerj.16781_

## Round 0.1 · original submission · Minor Revisions

This is an excellent manuscript. It requires several, but mostly minor revisions.

Requested revisions, additions:

a brief explanation why semilandmarks were not used for the curves, in Methods

include the rationale why the two phylogeny inference methods (ML and BI) were summarized together, in Methods

include an additional references that support your criterion for choosing the genetic distance for differentiating species. Regarding this figure, you should include a note in the legend, indicating clade B represents D. rotundus n. sp. Additionally, in line 1084 where it says "populations are constant..." you should refer to this figure as it follows "populations (clades A and B, Fig. 15)..."
see line 393: add to the Method section that expansions of palps of type specimens is usually not permitted, but if multiple specimens are available, ask the curator of the collection for permission.

Add to Fig.4C caption, if figure is based on expanded palp,
This editor sees no reason to describe the species from Reunion in this manuscript

Format change request: Material examined section: All material examined should be listed together. For kalanaro, the kalanaro material is listed (correctly) under Material examined starting at Line 589. Under Remarks, Line 671: material and types of straeleni is mentioned. Instead, add the examined straeleni material at a separate line after 595, e.g.,D. straeleni with the label information, in the same format as for the kalanaro material

Line 40: “Of raft spiders, genus Dolomedes Latreille, 1804,
literature only reports a single species on Madagascar”. Please correct, two species were reported, see WWSC.

Line 402: please cite Sierwald & Coddington: documentation of RTA function during palp expansion in Dolomedes: Sierwald, P. & Coddington, J. A. (1988). Functional aspects of the male palpal organ in Dolomedes tenebrosus, with notes on the mating behavior (Araneae, Pisauridae). Journal of Arachnology 16: 262-265.

Line 40: “Of raft spiders, genus Dolomedes Latreille, 1804”

Image question:

Were the habitus images in Figs… also taken with the Keyence microscope? If not, please indicate imaging technique/instrumentation. All images are very good.

Format change request: Material examined section: All material examined should be listed together. For kalanaro, the kalanaro material is listed (correctly) under Material examined starting at Line 589. Under Remarks, Line 671: material and types of straeleni is mentioned. Instead, add the examined straeleni material at a separate line after 595, e.g.,D. straeleni with the label information, in the same format as for the kalanaro material


Language
two reviewers provided excellent language advice. I checked all the changes and agree with each. Please make the changes


Additional language issues:
Line 293: 'of the Kalanoro groups', change to 'of the Kalanoro group.'
Line 490: check all spellings of ‘strait’ vs ‘straight’ you mean straight
Line 498: use ‘distinguished from’ instead of ‘diagnosed from’ throughout the manuscript
Line 1255: Journal of (m)Morphology

**Language Note:** The review process has identified that the English language must be improved. PeerJ can provide language editing services - please contact us at [email protected] for pricing (be sure to provide your manuscript number and title). Alternatively, you should make your own arrangements to improve the language quality and provide details in your response letter. – PeerJ Staff

Reviewer 1 ·

Basic reporting

The authors should do a final English editing to their text, there are some minor typos in the English writing, for example as it follows below:

335 … show higher variation compare to that of the other structures…
==== it should say “… compared to …” check it in the text and change it accordingly
484 Dolomedes can be diagnosed from Mangromedes by i) the strait and none pseudo-
===== change to “… can be distinguish from… by i) the straight and not pseudo…”
509 metatarsal spines that do not overlapping each other…
==== change to “… that do not overlap each…
553 … dorsal edge of LA with a strait section…
=== replace “strait” for “straight” and change it in the entire text as it corresponds
606 … Carapace pear-shaped, light brown bearded with
==== change to “… light brown with… Remove “bearded” from the species descriptions or replaced with “provided”
622 spots distribute on the dorsal abdomen …
==== it should say “… distributed on…
624 / 723 and others “… Palp tibia with a highly sclerotized RTA which divided into…
=== change to “… RTA divided into…
767 morph of is in general less abundant in the surveyed regions…
=== change to “morph is …
768 … coloration variations also present in female D. gregoric sp. nov
==== change to “… variations also occur in…
770 … species. Known inhabits only …
=== change to “… Known to inhabit…
773 … Population size much smaller than the coexisted D. bedjanic sp. nov.
======= change to “… much smaller than that of D. bedjanic sp. nov. with which it cohabits.

Experimental design

The authors mention in line 123 “original Madagascar sequences of a Dolomedes species … from Réunion”, I'm curious why did they not describe this new species in this work

Validity of the findings

No comment

Additional comments

Although your figure 15 summarize your molecular results very well, it could be better if you include in the methods section additional references that support your criterion for choosing the genetic distance for differentiating species. Regarding this figure, you should include a note in the legend, indicating clade B represents D. rotundus n. sp. Additionally, in line 1084 where it says "populations are constant..." you should refer to this figure as it follows "populations (clades A and B, Fig. 15)..."

·

Basic reporting

This is an excellent manuscript describing several new species of Dolomedes. Madagascar is a hyperdiverse country and I am not particularly surprised they've found congeners in sympatry. The methodology is sound, and the labelling of the palpal and epigyne structures is accurate. Plates are overall nicely presented and taxonomic characters are clear. All other figures are accurate and appropriate for the work. Overall, the changes needed are very minor. Most are grammatical corrections to improve the readability. I do also want to see it mentioned that whilst palp expansion is an important modern method, there are historical specimens for which this would not appropriate/may not be permitted by a museum. It is important to state this for the benefit of future workers so that this is understood by all, especially since the authors propose their framework be reproduced by others. Once revised, I think this manuscript is more than adequate for publication. Great work by the authors!

Specific details:

Line 164: Please cite Photoshop properly as done with other programs in the methodology.
Line 253: change 'under' to 'using'
Line 277: change 'they' to 'both sexes'
Line 293: change 'have narrower carapace' to 'have a narrower carapace'
Line 294: change 'than' to 'compared to'
Line 294: remove double space
Line 308: change 'strait' to 'straight'
Line 378: suggest changing 'discoveries' to 'research'
Line 381: change 'are' to 'were'
Line 389: change 'Based solely on very few characteristics when identifying Dolomedes can be risky' to 'Utilising only very few characteristics when identifying Dolomedes can be risky.'
Line 393: Whilst I agree this seems to be the best methodology, you should acknowledge the caveat that several Dolomedes species are known only from a single male specimen in historically-important museums. Palp expansion of type material is sometimes not appropriate for very fragile/bleached specimens. In such cases, topotypes would need to be collected in order to analyse expanded palps. This does not apply to you work but if you want this method to be reproduced by others, it would be beneficial to mention this issue and how it might be overcome.
Line 476: change 'The structure' to 'This structure'
Kine 484: change 'strait' to 'straight'
Line 484: change 'none' to 'non'
Line 485: change 'tarsus' to 'tarsi'
Line 490: change 'none' to 'non'
Line 491: change 'tarsus' to 'tarsi'
Line 490: change 'strait' to 'straight'
Line 499: add hyphen between high and contrast
Line 509: change 'overlapping' to 'overlap'
Line 526: add comma between indistinct and margins
Line 546: change 'that' to 'than'
Line 553: change 'strait' to 'straight'
Line 557: change 'strait' to 'straight'
Line 770: change 'Known inhabits' to 'Known to inhabit'
Line 773: change 'coexisted' to 'sympatric'
Line 876: change 'compare' to 'compared'
Line 889: change 'species groups' to '"Hydadtostella" species group'
Line 953: change 'strait' to 'straight'
Line 1095: change 'thanks the' to 'thanks to'
Caption of Fig. 19: change 'none' to 'non'

Experimental design

Good, no further comments.

Validity of the findings

Good, no further comments.

Additional comments

N/A

·

Basic reporting

The English is clear, I only detected two small mistakes:
Line 335. Replace compare by compared.
Line 431. Replace 20222 by 2022.

Experimental design

Line 183. I wonder why the authors did not use semilandmarks for the analysis of curved shapes such as the embolus. I would suggest briefly justifying why the followed method was preferred over the use of semilandmarks.
Line 233. Please specify which model was used for ML.
Line 240. As I am not familiar with this practice and other readers might not be as well, I would suggest including the rationale why the two phylogeny inference methods (ML and BI) were summarized together, specially taking into account that bootstrap support values and posterior probability do not give exactly the same information and might not be totally comparable.

Validity of the findings

no comment

Additional comments

This is an excellent article that uses an integrative approach to delimit species, combining morphology, genetics, ecology and distribution. By doing so, authors describe four new species of Dolomedes from Madagascar, and propose key diagnostic traits for future studies on this genus. I only have few comments that hopefully can help make some parts clearer in this article that is already of good quality.

---

## Round 0.2 · accepted · Accept

I reviewed the entire manuscript. In the revised version of the manuscript, all reviewers' and editor's requests have been fully integrated. The manuscript is ready for publication.